# In-situ spectroscopic probe of the intrinsic structure feature of single-atom center in electrochemical CO/CO₂ reduction to methanol

Xinyi Ren[1,2,10], Jian Zhao[1,10], Xuning Li [1] ✉, Junming Shao[3], Binbin Pan[4,5], Aude Salamé [3], Etienne Boutin[3], Thomas Groizard[3], Shifu Wang[1,6], Jie Ding [7], Xiong Zhang[1], Wen-Yang Huang[8], Wen-Jing Zeng[8], Chengyu Liu [3], Yanguang Li [4,5], Sung-Fu Hung [8] ✉, Yanqiang Huang [1], Marc Robert [3,9] ✉ & Bin Liu [7] ✉

While exploring the process of CO/CO₂ electroreduction (CO$_x$RR) is of great significance to achieve carbon recycling, deciphering reaction mechanisms so as to further design catalytic systems able to overcome sluggish kinetics remains challenging. In this work, a model single-Co-atom catalyst with well-defined coordination structure is developed and employed as a platform to unravel the underlying reaction mechanism of CO$_x$RR. The as-prepared single-Co-atom catalyst exhibits a maximum methanol Faradaic efficiency as high as 65% at 30 mA/cm² in a membrane electrode assembly electrolyzer, while on the contrary, the reduction pathway of CO₂ to methanol is strongly decreased in CO₂RR. In-situ X-ray absorption and Fourier-transform infrared spectroscopies point to a different adsorption configuration of *CO intermediate in CORR as compared to that in CO₂RR, with a weaker stretching vibration of the C−O bond in the former case. Theoretical calculations further evidence the low energy barrier for the formation of a H-CoPc-CO⁻ species, which is a critical factor in promoting the electrochemical reduction of CO to methanol.

As the cornerstone for sustainable production of fuels and chemicals, electrochemical CO/CO₂ reduction reaction (CO$_x$RR) has attracted increasing research interests due to its great promise in the mitigation of energy and environmental problems[1–8]. While CO₂RR has received more attention due to its potentiality to produce high value-added C$_{2+}$ products (e.g., ethylene, ethanol, and acetic acid) from renewable electricity and also because of its abundancy and low toxicity[9–17], the direct utilization of CO as a reactant can nevertheless increase the near-surface CO concentration and significantly suppress the competitive hydrogen evolution reaction (HER), thus greatly enhancing the catalytic performance for the selective production of desired fuels/chemicals[18–22]. However, the development of novel catalytic materials

[1]CAS Key Laboratory of Science and Technology on Applied Catalysis, Dalian Institute of Chemical Physics, Chinese Academy of Sciences, Dalian 116023, China. [2]University of Chinese Academy of Sciences, Beijing 100049, China. [3]Université Paris Cité, Laboratoire d'Electrochimie Moléculaire, CNRS, F-75006 Paris, France. [4]Institute of Functional Nano & Soft Materials (FUNSOM), Soochow University, Suzhou 215123, China. [5]Jiangsu Key Laboratory for Advanced Negative Carbon Technologies, Soochow University, Suzhou 215123, China. [6]Department of Chemical Physics, University of Science and Technology of China, Hefei 230026, China. [7]Department of Materials Science and Engineering, City University of Hong Kong, Hong Kong SAR 999077, China. [8]Department of Applied Chemistry, National Yang Ming Chiao Tung University, Hsinchu 300, Taiwan. [9]Institut Universitaire de France (IUF), F-75005 Paris, France. [10]These authors contributed equally: Xinyi Ren, Jian Zhao. ✉e-mail: lixn@dicp.ac.cn; sungfuhung@nycu.edu.tw; robert@u-paris.fr; bliu48@cityu.edu.hk

with simplified chemical protocols that would achieve highly efficient and robust $CO_x$ electroreduction is still a great challenge.

Selectivity to $C_2$ and $C_{2+}$ products is still unsatisfactory due to the involvement of complex reaction intermediates (i.e., *CO, *CHO, *COH, *OCCO, *OCCOH, etc.)[23–27]. Strikingly, it holds true also for $C_1$ products, with still a lack of catalysts for high-rate production of formaldehyde and methanol with high selectivity. Among the intermediates mentioned above, *CO has been identified as the key reaction intermediate for $C_1$, $C_2$, and $C_{2+}$ product formation in $CO_2RR$[8,28–32]. Thus, the development of CORR with high activity and selectivity for various products (e.g., methanol) could be an important step toward the development of a high-efficient tandem catalyst for $CO_2RR$[33–40]. The above considerations point toward the importance of unraveling the mechanisms of the CORR so as to progress toward a rational design of effective catalysts. Otherwise, the intrinsic factors driving the CORR and the structural features of *CO intermediates have yet to receive little attention.

Single-atom catalysts with well-defined structures allow us to gain atomic-scale insight into the underlying reaction mechanism and identify the real structure of catalytic intermediate in $CO_xRR$. To date, metal−nitrogen−carbon materials (M−N−C) have been widely investigated in $CO_xRR$. An alternative approach is the use of molecular catalysts being adsorbed or grafted at conductive supports, such as carbon particles, nanotubes, graphene, etc. Molecular catalysts allow for tuning the environment and electronic properties at the catalytic site with unrivaled precision, such properties being additionally modulated by the interaction with the conductive support[41]. Metal porphyrins, corroles (MCs), phthalocyanines (MPcs) as well as various complexes have been widely employed for $CO_xRR$[42–44]. Apart from CO and $HCOO^-$ as the most commonly observed products from $CO_2RR$, recent studies also demonstrated the possible generation of methanol at the Co atom site (phthalocyanine adsorbed at carbon electrodes)[45,46]. Although some progresses have been made to elucidate CO conversion to methanol[47], most of the mechanistic aspects of methanol formation from CO or $CO_2$ electroreduction remain to be uncovered. Atomic insights into the exact structure of intermediates at the catalytic site are especially needed.

Herein, by anchoring cobalt phthalocyanine on multi-walled carbon nanotubes, a model single-Co-atom catalyst was developed to decipher the structural features of the active species in $CO_xRR$ to produce methanol and the underlying mechanism. A methanol Faradaic efficiency (FE) is as high as 47.8% was realized at −0.70 V vs. reversible hydrogen electrode (RHE) in an H-cell during CORR (0.5 M $K_2HPO_4$ electrolyte, pH = 9.6), while surprisingly, the methanol production was essentially suppressed in $CO_2RR$. An optimal methanol FE of 65% was further achieved in a membrane electrode assembly (MEA) type electrochemical cell. Combined with advanced in situ spectroscopic measurements and theoretical calculations, key intermediates related to CO bound to the cobalt atom (*CO intermediates) have been identified through in situ IR spectroscopy and additional insights from theoretical calculations, thus revealing the fundamental reasons for the electrochemical reduction of CO (instead of $CO_2$) to methanol.

## Results and discussion

The outlined preparation procedure took advantage of π−π interactions between the aromatic phthalocyanine ring in the CoPc and carbon matrix in the MWCNTs (CoPc/MWCNT) by means of readily mixing the cobalt complex and MWCNTs in N,N-dimethylformamide (DMF) (as detailed in the "Methods"). As shown in Fig. 1a and Supplementary Fig. 1, the morphology of the resulting CoPc/MWCNT well inherited the original structure of MWCNTs with an average diameter of ~8 nm, and no aggregation of large CoPc clusters was detected, which is consistent with the X-ray diffraction (XRD) measurements (Supplementary Fig. 2a). Energy dispersive X-ray spectroscopy (EDX) mappings (Fig. 1b) revealed the homogeneous distribution of N and Co elements throughout the carbon support, verifying a good dispersion of CoPc on MWCNTs. The aberration-corrected high-angle annular dark-field scanning transmission electron microscopy (HAADF-STEM) image further revealed that the isolated Co atoms were uniformly anchored on the surface of MWCNTs (Fig. 1c).

To gain information about the chemical environment and electronic structure of Co-atom in CoPc/MWCNT, X-ray absorption spectroscopy (XAS) measurements were performed. As shown in Fig. 1d, CoPc/MWCNT exhibited a similar $D_{4h}$ symmetry structure and cobalt

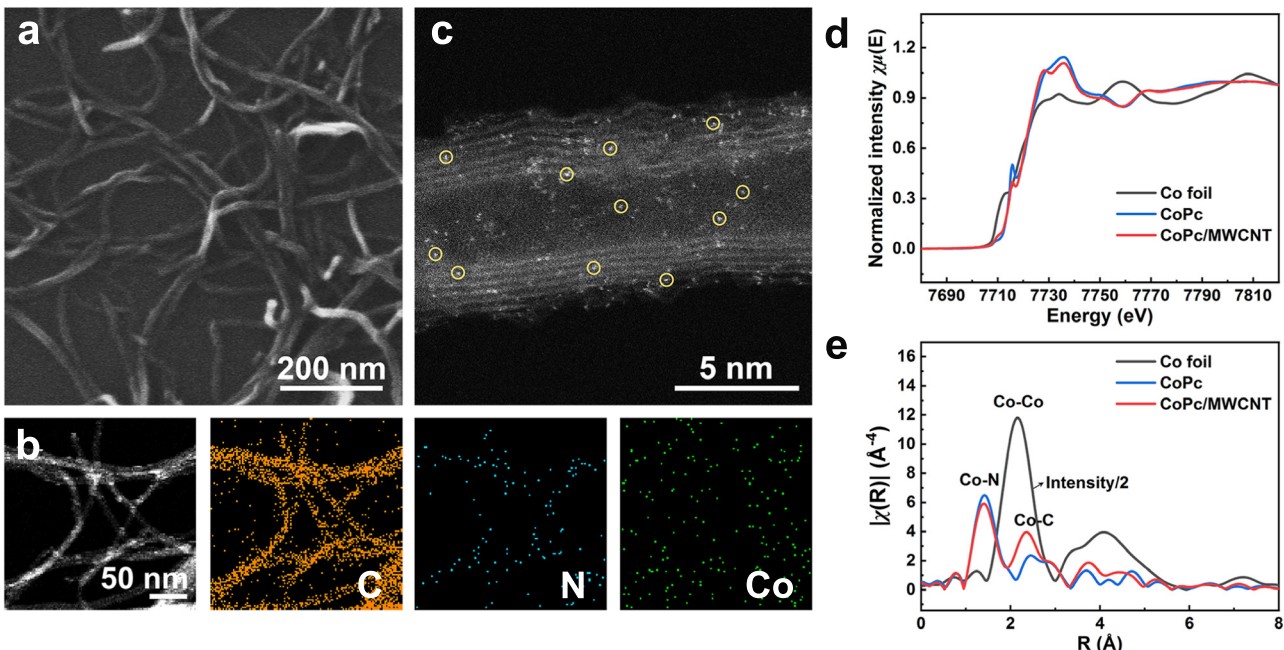

**Fig. 1 | Structural characterization of the as-prepared CoPc/MWCNT. a** FESEM image of CoPc/MWCNT. **b** STEM image and the corresponding EDX elemental mapping [C (orange), N (blue), and Co (green)] of CoPc/MWCNT. **c** HAADF-STEM image of CoPc/MWCNT. The circled bright spots highlight the dispersed Co atoms. **d** Co K-edge XANES spectra and **e** EXAFS spectra in R space of Co foil, CoPc, and CoPc/MWCNT.

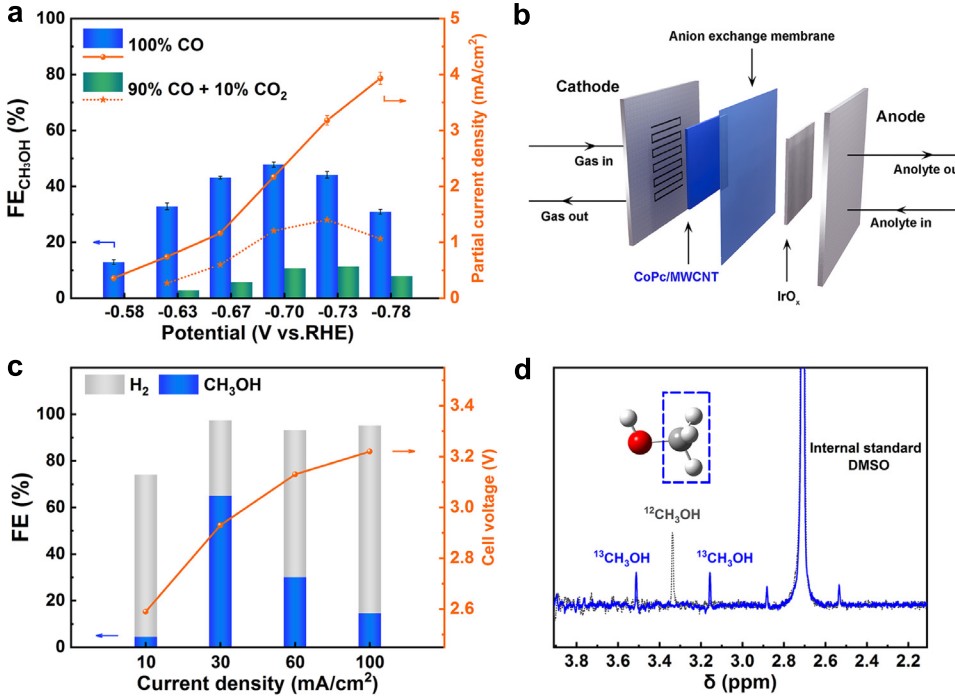

**Fig. 2 | Electrochemical CORR. a** $FE_{CH_3OH}$ under 100% CO (blue bar), 90% CO + 10% $CO_2$ (green bar), and $CH_3OH$ partial current density under 100% CO (orange solid line), 90% CO + 10% $CO_2$ (orange dotted line), of CoPc/MWCNT at various applied potentials. Error bars represent the standard deviation of three independent measurements. **b** Schematic illustration of the MEA cell. **c** $FE_{CH_3OH}$ and $FE_{H_2}$ at different current densities in 0.25 M $K_2HPO_4$ electrolytes in MEA cells. **d** $^1H$ NMR spectra of the electrolyte after electrolysis ($E = -0.70$ V vs. RHE, $t = 1$ h) in $^{12}CO$- (gray trace) and $^{13}CO$-saturated (blue trace) solutions.

oxidation state ($Co^{2+}$) to that of CoPc with the same $1s \rightarrow 4p_z$ transition at 7715.1 eV[48,49], which is consistent with the X-ray photoelectron spectroscopy (XPS) measurements (Supplementary Fig. 2b). In addition, a main peak at ca. 1.4 Å that could be assigned to the Co−N bond in the first coordination shell was observed in the Fourier-transformed $k^3$-weighted extended X-ray absorption fine structure (FT-EXAFS) spectrum (Fig. 1e). On the other hand, no obvious peaks related to Co−Co coordination could be observed, suggesting the successful anchoring of the complex on MWCNTs without aggregation. The distance of C (marked as Co−C) at the second coordination sphere reduced from 2.44 to 2.35 Å, indicating a distortion of the phthalocyanine molecule due to the interaction with the nanotube.

The electrochemical CORR performance was first evaluated by LSV on a rotating disk electrode (RDE) in 0.5 M $K_2HPO_4$ electrolyte. As shown in Supplementary Fig. 3a, the onset potential is more positive in the CO-saturated condition than that in the Ar-saturated solution. A cathodic peak centered at ca. −0.53 V vs. RHE associated with CO reduction was observed, indicating the high CO electroreduction activity of CoPc/MWCNT. The time-dependent total current density of CORR over CoPc/MWCNT was further recorded in an H-cell. As shown in Supplementary Fig. 3b, stable current densities could be continuously collected for each of the applied potentials during 1 h. Proton nuclear magnetic resonance ($^1H$ NMR) measurements showed that methanol was the only liquid product (Supplementary Fig. 4), and the total FE of $CH_3OH$ plus $H_2$ was found to be close to 100% over the entire tested potential window (Supplementary Fig. 5). Noteworthy, $FE_{CH_3OH}$ reached a maximum value of 47.8% at −0.70 V vs. RHE along with a partial current density ($j_{CH_3OH}$) of ca. 2.2 mA/cm$^2$ (Fig. 2a). The complex was further deposited at a gas diffusion electrode and inserted in an MEA (Fig. 2b and Supplementary Fig. 6). With optimization of the cell, a CoPc/MWCNT (0.3 mg/cm$^2$) loaded gas diffusion layer (GDL) was assembled to the cathode and tested with a CO gas at a flow rate of 40 SCCM, whereas the anode was fed with recirculated 0.25 M $K_2HPO_4$ aqueous electrolyte at a flow rate of 10 mL/min. The $CH_3OH$ selectivity

was further increased to as high as 65% at 30 mA/cm$^2$ ($\Delta E_{cell} = 2.93$ V), a partial current density $j_{CH_3OH}$ of 19.5 mA/cm$^2$ (Fig. 2c). These results illustrate the potential of CoPc/MWCNT as the catalyst for high-rate production of methanol from CORR, with yet ample room for future improvement.

The isotope labeling experiment, conducted in a $K_2HPO_4$ solution saturated with $^{13}CO$, gave a complete splitting of the $^1H$ NMR singlet peak at $\delta = 3.34$ ppm ($CH_3$ protons) to a doublet peak ($j_{C-H} = 142$ Hz)[50], demonstrating that CO is the source for the observed methanol production (Fig. 2d). Moreover, no $CH_3OH$ could be detected in similar experiments where CoPc/MWCNT was replaced by metal-free $H_2Pc$ molecule adsorbed on MWCNTs ($H_2Pc$/MWCNT) and single-Co-atom catalyst derived from calcination (Supplementary Fig. 7). It suggests that the well-defined coordination environment of the Co atom in CoPc molecule should be responsible for the evolution of methanol from CORR. As displayed in Supplementary Fig. 8, the $FE_{CH_3OH}$ decreased by only 2.8% after 8 h of continuous CORR at −0.70 V vs. RHE, demonstrating the high catalytic stability of CoPc/MWCNT in CORR.

An experiment with a gas mixture consisting of 90% CO and 10% $CO_2$ as the reactant was also performed. Surprisingly, methanol formation was suppressed to a large extent (maximum $FE_{CH_3OH}$ of ca. 10%, see the calculation of FE in the Supplementary Information) along with a significantly reduced $j_{CH_3OH}$ (Fig. 2a and Supplementary Fig. 9). Given that the reactivity for methanol formation is obviously not correlated to the CO reactant concentration (the estimated $FE_{CH_3OH}$ would be much higher if the CORR activity was solely contributed by CO composition, i.e., $FE_{CH_3OH} = 43.02\%$ with 90% CO in the inlet gas), these results suggest competitive adsorption of CO and $CO_2$ at the Co atom with a stronger $CO_2$ binding and in addition it also suggests that $CO_2$ reduction mainly leads to CO. An additional experiment was designed upon preparing an almost equimolar amount of $^{12}CO$ and $^{13}CO_2$ as reactants (from a gas mixture containing 96.5% $^{12}CO$ and 3.5% $^{13}CO_2$, which corresponds to concentrations of 0.95 mM and 1.15 mM

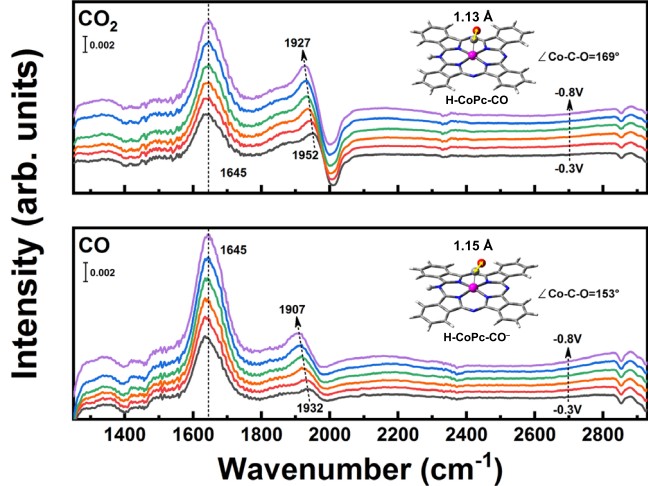

**Fig. 3 | Identification of reaction intermediates.** In situ, FTIR spectra of CoPc/MWCNT measured at $E = -0.3$ V – $-0.8$ V vs. RHE in $CO_2-$ (top) and CO-saturated (bottom) electrolytes. The inset in each figure shows the DFT-optimized structures of H-CoPc-CO (top) and H-CoPc-CO$^-$ (bottom); the C–O bond length and Co–C–O bond angle are also indicated.

respectively for the gases, Supplementary Fig. 10). A short time CPE (10 min) was then performed at $E = -0.70$ V vs. RHE. $^1$H NMR analysis of the electrolyzed solution indicated that the produced methanol was exclusively issued from the non-labeled carbon monoxide reduction since only the singlet peak of $^{12}CH_3OH$ was observed. The yield for methanol production was reduced to ca. 10% FE as compared to electrolysis with CO as the only reactant. These experiments demonstrate that (a) there is a competition between $CO_2$ and CO for binding to CoPc/MWCNT; (b) this competition is in favor of $CO_2$; and (c) the CO formed upon $CO_2$ reduction is preferentially desorbed from the metal center rather than reduced to methanol. This also explains why methanol formation is essentially suppressed under a $CO_2$ atmosphere, with only CO as the main product and $H_2$ as a by-product, as reported in most studies. Indeed, as initially observed by Boutin et al., only a small amount of methanol could be detected during $CO_2$RR at CoPc/MWCNT (typically 0.3% FE at pH 7.2 in 0.1 M $KHCO_3$ electrolyte)[45]. Upon extensive efforts to optimize the catalyst through ink formulation, the FE for $CH_3OH$ production was increased to a maximum value of ca. 8% (Supplementary Fig. 11), further confirming that $CO_2$ is not a good reactant to generate methanol with CoPc/MWCNT as a catalyst.

To investigate the mechanistic origin of methanol formation from CO electroreduction in situ, Raman spectroscopy experiments were conducted on CoPc/MWCNT (Supplementary Figs. 12–14). As shown in Supplementary Fig. 12, at open circuit potential (OCP), the in situ Raman spectra presented a characteristic peak at 750 cm$^{-1}$, which could be assigned to the stretching vibration of M–N bonds and C–N–C bridge bonds[51]. At −0.7 V vs. RHE, the peak intensity at 750 cm$^{-1}$ significantly dropped along with the appearance of a shoulder peak at 745 cm$^{-1}$ in both CO$^-$ and $CO_2$-saturated electrolytes, illustrating that the electroreduction process occurred on CoPc, which altered the Co–N coordination environment and the surrounding C–N–C configuration. Furthermore, negligible change over $H_2$Pc/MWCNT was observed across the entire potential range (Supplementary Fig. 13), indicating that the CO$_x$RR did not proceed on the Pc ring but rather on the Co-active sites.

In situ FTIR measurements were further conducted to gain insights into the reaction intermediates involved in the CO$_x$RR. As shown in Fig. 3 and Supplementary Fig. 15, a broad peak located at 1645 cm$^{-1}$ associated with the O–H bending mode of water molecules remained unchanged for both reactions. The peak at 1952 cm$^{-1}$, which

could be assigned to the adsorption of CO on active sites upon a linear Co–CO configuration[52], was gradually shifted to 1927 cm$^{-1}$ upon applying more cathodic potential in the presence of $CO_2$, which is due to the Stark effect arising from the decreased resonant frequencies in negative electric fields[53]. A similar change was also observed in the CO-saturated solution. However, the peak due to CO being adsorbed at CoPc/MWCNT in CO-saturated solution was located at lower wavenumbers (by 20 cm$^{-1}$) as compared to that in $CO_2$-saturated solution, indicating the generation of weakened C–O bond in the *CO intermediate on the Co site during CORR.

To further reveal the electronic properties of the single-Co-atom sites in CO$_x$RR, in situ, XAS measurements were performed in CO$^-$ and $CO_2$-saturated electrolytes. Figure 4 shows the normalized Co K-edge X-ray absorption near edge spectroscopy (XANES) spectra, with two dipole-allowed transitions being clearly observed in the rising edge: a sharp peak at ~7715 eV corresponding to $1s \rightarrow 4p_z$ transition reflecting the $D_{4h}$ planar symmetry of CoPc molecule and a white-line transition at ~7735 eV corresponding to $1s \rightarrow 4p_{x,y}$ transition. When cathodic potentials of −0.5 V and −0.7 V vs. RHE were applied, the intensity of $1s \rightarrow 4p_z$ electron transition peak gradually decreased (Fig. 4a, b), suggesting that the change of coordination structure around Co modified the original $D_{4h}$ planar symmetric geometry. Additionally, the absorption edge shifted toward lower energies, and the white-line intensity decreased, indicating the reduction of the Co average oxidation state during electroreduction. Figure 4c–e compares the Co K-edge XANES spectra in $CO_2$ and CO atmospheres. As can be observed, with potential being decreased from $E = -0.5$ V to −0.7 V vs. RHE, the drops in white-line intensities were more pronounced in the CO atmosphere, indicating that Co sites exhibit a lower oxidation state in CORR compared to that in $CO_2$RR.

DFT calculations were performed to gain a theoretical understanding of the reaction pathway for methanol formation over a single-Co-atom site. As shown in Supplementary Fig. 16, two redox peaks of CoPc/MWCNT appeared under the Ar atmosphere. The first reversible peak at around 0.20 V vs. RHE arises from the reduction of Co(II) to Co(I), while the second reduction peak at around −0.28 V vs. RHE corresponds to the delocalization of the charge obtained onto the macrocycle[48,54], resulting in the protonation at the ring. The calculation results of the relative single point energy show that the CoPc is easily protonated at the ligand (denoted as H-CoPc), making it more thermodynamically stable (Supplementary Fig. 17), which is also confirmed by the fact that the peak at 750 cm$^{-1}$ in the in situ Raman spectra cannot be fully recovered upon setting back the electrode to OCP, while fully recovered after calcination at 300 °C under Ar atmosphere (Supplementary Fig. 18). Therefore, H-CoPc was proposed as starting species for CO/$CO_2$ electrochemical reduction and more in-depth understanding on the reaction route could be supported by DFT calculations.

The DFT-optimized structure models of H-CoPc and corresponding intermediates are shown in Supplementary Figs. 19 and 20. On the basis of the reaction Gibbs free energies (Fig. 5a), the rate-limiting step in $CO_2$RR is CO desorption from H-CoPc-CO species (0.38 eV), while in CORR, it is the protonation of H-CoPc-CO$^-$ (0.40 eV). Additionally, a quite high energy barrier of 0.72 eV is required for the reductive conversion of H-CoPc-CO into H-CoPc-CO$^-$ intermediate, meaning that both H-CoPc-CO and H-CoPc-CO$^-$ would accumulate in the solution and thus could be detected by in situ FTIR and XAS measurements. The optimized structures of H-CoPc-CO and H-CoPc-CO$^-$ are shown in the inset of Fig. 3, where *CO binds to Co at different bond angles, and the calculated C–O bond length in H-CoPc-CO$^-$ is 1.15 Å, longer than that in H-CoPc-CO (1.13 Å). Besides, the predicted IR vibrational frequencies were also in good agreement with the shifting trend observed in in situ FTIR (Supplementary Fig. 21). The Bader charge analysis gives that the charge of Co atom in H-CoPc-CO and H-CoPc-CO$^-$ is 1.18 e and 1.03 e, respectively (labeled on Fig. 4f and

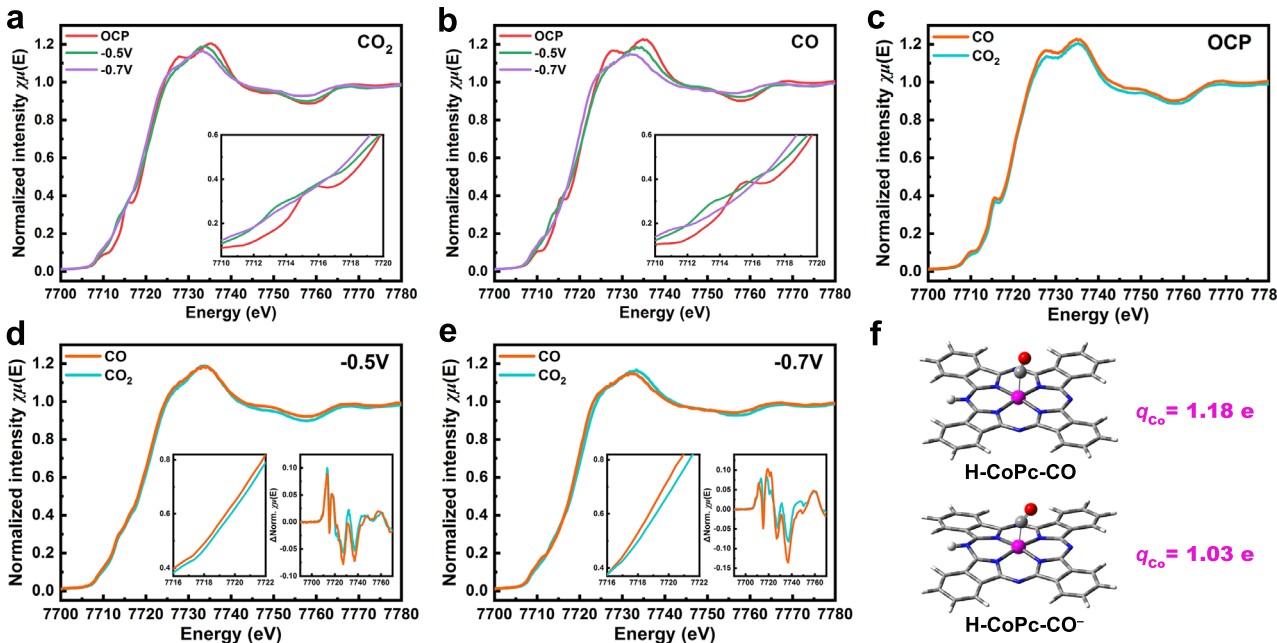

**Fig. 4 | In-situ XAS characterization of CoPc/MWCNT.** Normalized Co K-edge XANES spectra of CoPc/MWCNT recorded at OCP, $E = -0.5$ V and $-0.7$ V vs. RHE in **a** $CO_2$ and **b** CO-saturated electrolyte. Comparison of Co K-edge XANES spectra in $CO_2$ and CO atmospheres at **c** OCP, **d** $E = -0.5$ V vs. RHE and **e** $E = -0.7$ V vs. RHE. The inset in each panel is the zoomed-in region showing the near-edge shift (left) and the differential spectra (right). **f** Bader charges of Co ion in H-CoPc-CO and H-CoPc-CO⁻ species.

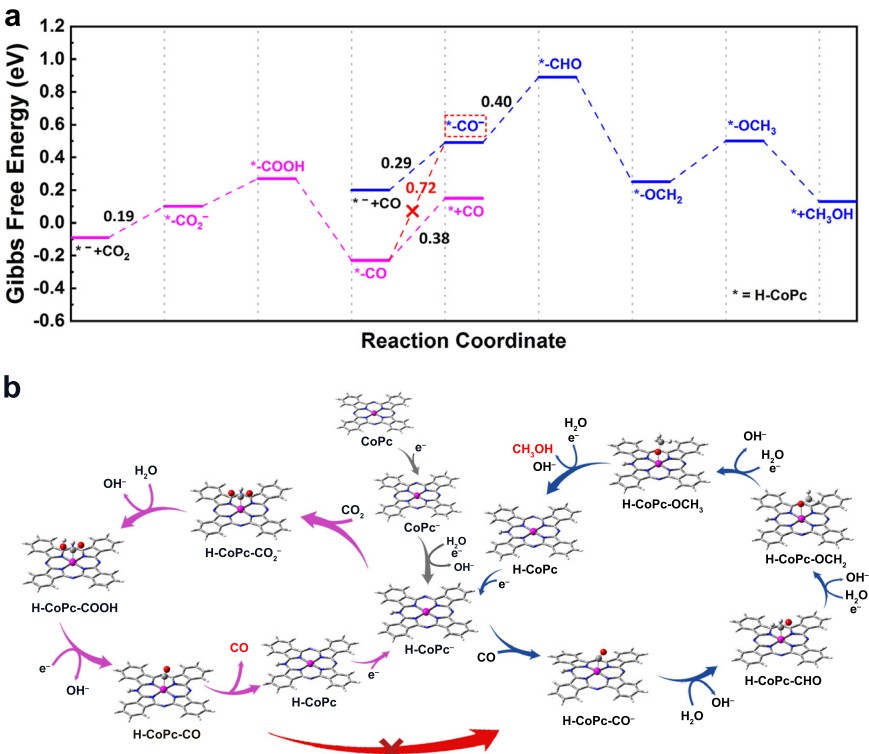

**Fig. 5 | Theoretical calculations. a** Calculated Gibbs free energy diagrams, where * stands for the protonated CoPc (H-CoPc). **b** Proposed reaction steps for the conversion of $CO_2$ to CO, CO to $CH_3OH$ on CoPc/MWCNT.

Supplementary Table 1), further showing the higher electron density surrounding the Co atom in H-CoPc-CO⁻ as compared to H-CoPc-CO.

As shown in Fig. 5a, the $CO_2$-to-CO pathway is more energetically favorable. Compared to the adsorption of CO (0.29 eV), the lower reaction energy barrier for $CO_2$ adsorption on H-CoPc⁻ (0.19 eV) results in the preferential adsorption and activation of $CO_2$ even with a small amount of $CO_2$ in a CO-saturated electrolyte. Furthermore, the low energy barrier (0.38 eV) for CO desorption makes CO the primary reduction product in $CO_2$RR. Therefore, the methanol pathway is largely suppressed in the CO/$CO_2$ mixtures. On the contrary, in CORR, the

energy barrier for the formation of H·CoPc-CO⁻ intermediate is much lower (0.29 eV, to be compared to 0.72 eV for the reductive conversion of H·CoPc-CO into H·CoPc-CO⁻), indicating that CO can be more easily activated on H·CoPc⁻ along the CO-to-CH₃OH pathway, while it is challenging to achieve such activation in the CO₂-to-CO pathway because of the very unlikely possibility that H·CoPc-CO can convert to H·CoPc-CO⁻ upon reduction. Additional DFT calculations further support that the reaction path of CO-to-CHO is more preferred compared to the CO-to-COH path (Supplementary Figs. 22 and 23), and the whole methanol pathway involves the consecutive reduction of *CHO, *OCH₂, and *OCH₃ intermediates. In addition, another metal phthalocyanine, including FePc, NiPc, and CuPc, did not exhibit any CORR activity toward methanol (Supplementary Fig. 24), which is most probably due to their higher energy barriers for the CO-to-CHO path compared to CoPc (Supplementary Fig. 25). Therefore, the direct involvement of CO as the reactant can lower the reaction energy barrier for the formation of H·CoPc-CO⁻ intermediate, leading to the high methanol selectivity in CORR on single-Co-atom catalytic site.

In summary, taking advantage of combined electrochemical studies, in-situ spectroscopic measurements, and theoretical calculations on a model single-Co-atom molecular catalyst, we have deciphered important mechanistic aspects of the electrochemical CO₂/CO reduction for methanol formation. The intrinsic structural feature of a single-Co-atom center associated with different configurations of *CO intermediate in CO₂RR and CORR has been evidenced. In CORR, a higher electron density of a single-Co-atom center was detected by in-situ XAS measurements and proved to be responsible for a configuration of *CO with weaker stretching vibration of the C−O bond, which could effectively reduce the energy barrier for the formation of H·CoPc-CO⁻ as the key catalytic intermediate, followed by further hydrogenation of *CO to *CHO, thus facilitating the methanol formation. In contrast, during CO₂RR, the preferential CO desorption hindered the further reduction of *CO, mainly suppressing the production of methanol. This study provides comprehensive insights into CO₂-to-CO and CO-to-CH₃OH electrochemical reaction pathways, which should pave the way for the future design of highly efficient catalysts for CO₂/CO electroreduction. Finally, it may be noted that the electronic property of the single Co-atom site is exquisitely dependent on the ligand structure and also on the conductive support (MWCNT in our case), it would be interesting to explore the effect of the carbon support on the activity for CO₂RR and CORR, respectively.

## Methods

### Preparation of CoPc/MWCNT catalyst
In a typical process for the synthesis of CoPc/MWCNT catalyst, 100 mg of MWCNTs were dispersed in 100 mL of DMF and sonicated for 1 h. Then, 10 mg of CoPc dissolved in DMF was added into the MWCNT suspension, followed by 30 min of sonication to obtain a well-mixed suspension. Subsequently, the mixed suspension was stirred at room temperature for 48 h. After collection by vacuum filtration and extensive washing with DMF and ethanol, the precipitate was transferred to a 60 °C oven to dry overnight to give the desired CoPc/MWCNT catalyst. H₂Pc/MWCNT was prepared using the same method, except CoPc was replaced by H₂Pc.

### Chemicals and materials
Cobalt phthalocyanine (CoPc) was purchased from Alfa Aesar Co. Ltd., Shanghai, China. Hydroxy multi-walled carbon nanotubes (−OH, 5.58 wt%) was gained from 3A Chemical Reagent Co. Ltd., China. Potassium phosphate dibasic anhydrous (K₂HPO₄, ≥99.0%) was purchased from Aladdin Co. Ltd., Shanghai, China. Analytical reagent (AR) grade N,N-Dimethylformamide (DMF) was obtained from Aladdin Co. Ltd., Shanghai, China. AR grade for other reagents used was obtained from Tianjin Chemical Reagent Co. Ltd. All chemicals mentioned above were used without further purification. An 18.2 MΩ cm resistivity of

ultrapure water (Millipore Milli-Q grade) was used to prepare all solutions. AvCarb Grade-P75T carbon fiber paper treated with 10 wt% PTFE was purchased from Fuel Cell Store.

### Preparation of GDL
Multi-walled carbon nanotubes (MWCNT, 6−9 nm diameter, 5 μm length, >95% Carbon) (20 mg) and CoPc (2 mg) were dispersed in 20 mL ethanol and sonicated for 30 min. Then, 200 μL 5 wt% Nafion solution (perfluorinated ion exchange Nafion© powder, in low aliphatic alcohol/H₂O) was added to the suspension. A well-mixed suspension was yielded by sonicating the mixture for another 30 min and further stirred constantly at room temperature overnight to obtain the final catalyst ink. To create the GDL loaded with CoPc/MWCNT, the catalyst ink was drop-casted onto carbon paper using a PTFE mask. The mask had a 1 cm² surface square hole with 3 mm thickness, which allowed it to receive approximately 100 μL of catalyst ink each time. After repeating the drop-casting 20 times, the ink-deposited carbon paper was placed at room temperature and dried for 3 days in air to form the final 1 cm² GDL.

### Preparation of single-Co-atom catalyst
The single-Co-atom catalyst was prepared according to the method of Yang et al.[55], 8 g melamine (C₃H₆N₆), 1.5 g ʟ-alanine (C₃H₇NO₂) and 0.13 g cobalt acetate tetrahydrate (Co(CH₃COO)₂·4H₂O) were mixed into a homogeneous precursor through ball-milling and then pyrolyzed in two stages: 600 °C for 2 h and 900 °C for 1 h in Ar atmosphere. Once cooled to room temperature, the product underwent leaching in 1 M HCl at 80 °C for 12 h and 1 M HNO₃ for 24 h, which effectively eliminated metal particles and unstable species. Subsequently, the powder was subjected to calcination at 850 °C in an Ar environment for 1 h to restore the crystalline structure of carbon.

### Calculation of FE
The FE of CH₃OH and H₂ was calculated by:

$$FE_i = \frac{Q_i}{Q_{\text{total}}} = \frac{N_i*n*F}{Q_{\text{total}}}$$

where $Q_i$ represents the charge employed in the reduction of a particular product. $Q_{\text{total}}$ is the total charge that has been transferred, and the unit is C. $N_i$ signifies the number of moles of a specific product, measured through GC or NMR and given in mol. $n$ denotes the number of electrons involved in the formation of the product, which is 4 in the case of CO transforming into CH₃OH during CORR because CO₂ to CH₃OH is almost completely inhibited at −0.63-−0.78 V vs. RHE. $F$ is the Faraday constant, whose value is 96485 C/mol.

### Calculation of partial current density
The partial current density was obtained upon scaling the total current density:

$$j_i = FE_i*j_{\text{total}}$$

$j_{\text{total}}$: total current density, mA/cm².

### Characterization
Elemental analysis was conducted by inductively coupled plasma optical emission spectroscopy. The cobalt content in CoPc/MWCNT is 0.85 wt%. The morphological information of the CoPc/MWCNT was examined with field-emission scanning electron microscopy (JSM-7800F) with an accelerating voltage of 5.0 kV. HAADF-STEM and the corresponding EDX mapping were conducted on a JEM-ARM200F STEM/TEM. XRD was acquired by a PAN Analytical Empyrean diffractometer with Cu Kα radiation at 40 kV and 40 mA. XPS measurements were carried out on a Thermofisher Escalab 250 Xi+

spectrometer using an Al Kα X-ray source with a pass energy of 30.0 eV. The 284.6 eV of the C 1$s$ peak was used to calibrate the binding energies. XAS, including XANES and EXAFS of Co K-edge, were carried out at the BL11B beamline at the Shanghai Synchrotron Radiation Facility, China. The energy was calibrated according to the absorption edge of pure Co foil. The acquired EXAFS data were processed according to the standard procedures using the ATHENA module of Demeter software packages. The Hanning window was utilized for the Fourier-transform of the $k^3$-weighted $\chi(k)$ data in the $k$-space ranging from 3.0 to 10.6 to real (R) space.

## In situ electrochemical measurements
During in situ Raman measurements, a customized electrochemical cell filled with 0.5 M $K_2HPO_4$ solution was used, and $CO_2$/CO flowed until saturation through the gas flowmeter. Raman spectra were collected using a Renishaw inVia microprobe Raman spectrometer, employing an excitation laser at 785 nm with 2.5 mW between 200 and 1800 cm$^{-1}$. An Ag/AgCl electrode and a Pt wire were used as the reference and counter electrode, respectively. A cathodic scan from −0.5 V to −0.8 V vs. RHE was applied to the working electrode for 120 s, and then the spectra were collected.

In situ electrochemical attenuated total reflection surface-enhanced infrared absorption spectroscopy experiments were conducted on a Nicolet iS50 FTIR spectrometer. The electrocatalyst was dropped onto an Au film to serve as the working electrode. A graphite rod and an Ag/AgCl electrode were used as counter and reference electrodes in all tests, respectively. The background was taken at OCP in Ar-saturated 0.5 M $K_2HPO_4$ electrolyte[56], and then the reference spectra under Ar atmosphere were collected from −0.3 V to −0.8 V vs. RHE with a step width and duration of 100 mV and 180 s, respectively. Afterward, CO/$CO_2$ gas flow was purged into the electrolyte until saturation. The background was collected again at OCP, and the spectra were then collected from −0.3 V to −0.8 V vs. RHE.

The in situ XAS analysis was conducted in the BL17C beamline in NSRRC (National Synchrotron Radiation Research Center, Taiwan). In a homemade electrochemical cell, a $Hg/Hg_2Cl_2$ electrode was selected as the reference electrode and a Pt foil as the counter electrode. The working electrode was prepared by dropping the catalyst ink containing 1 mg of catalyst onto one side of the 1 cm$^2$ carbon paper. Kapton tape was used to seal the cell with an X-ray passing through it. During electrolysis, the $CO_2$/CO was bubbled into the cell filled with 0.5 M $K_2HPO_4$ successively at a constant potential. Each potential was held for 10 min before measurement and for an additional 15 min using the chronoamperometry technique. The XAS raw data were background-subtracted and normalized. The differential spectra were obtained by subtracting the spectra obtained at OCP from the counterparts at −0.5 V and −0.7 V vs. RHE, respectively.

## Electrochemical measurements
CO electroreduction experiments were carried out in a customized three-compartment cell separated by Nafion 117 membrane. A saturated calomel electrode (SCE) ($Hg/Hg_2Cl_2$) and a platinum plate were used as the reference electrode and counter electrode, respectively. The working electrode was prepared by dispersing 0.5 mg of CoPc/MWCNT catalyst in 960 µL of water-isopropyl alcohol solution (volume ratio of 1:1) with 40 µL Nafion. The catalyst ink was then painted onto carbon paper (area = 1 cm$^2$). Chronoamperometry analysis was performed on a CHI660E potentiostat in 35 mL 0.5 M $K_2HPO_4$ electrolyte, and the current–time response was recorded in 3600 s with a sampling time interval of 0.128 s at each potential. For all the measurements, CO (with 1% Ar) was continuously purged into the solution at a flow rate of 20 mL/min (in the comparative experiments, 90% CO mixed with 10% $CO_2$ gas or 100%

$CO_2$ gas was purged). The linear sweep voltammograms (LSVs) were recorded on a rotating disc electrode in the potential range from −0.6 to −1.6 V vs. SCE with a scan rate of 5 mV/s. Cyclic voltammetry (CV) curves were recorded at a scan rate of 50 mV/s. All potentials were measured vs. SCE and were converted vs. RHE according to the Nernst equation: $E$ (vs. RHE) = $E$ (vs. SCE) + 0.2415 V + 0.0592 × pH. Gas-phase products were quantified by an online gas chromatograph (Agilent 7890B) equipped with a thermal conductivity detector and flame ionization detector. $^1$H-NMR spectra of the liquid products were recorded on a Bruker Advance 400 spectrometer via water suppression. 1 mL of electrolyte was mixed with 0.1 mL of $D_2O$ (containing 2.5 µL dimethyl sulfoxide as the internal standard). Quantification was made relative to the DMSO peak.

## CORR MEA electrolysis
The measurements were carried out in a 5 cm$^2$ MEA instrument (manufactured by Dioxide Materials Company), which consists of a titanium anode (serpentine flow field), the anode catalyst, ion-exchange membrane, the cathode catalyst and a stainless cathode plate (serpentine flow field). To prepare the cathode catalyst, 4 mg of CoPc/MWCNT catalyst powder was first dispersed in 4 mL of isopropanol and 4 mL of ultra-pure water with 30 µL of 5 wt% Nafion solution and subjected to vigorous ultrasonication for about 4 h. This catalyst ink was then sprayed onto a 2.5 × 2.5 cm$^2$ carbon fiber paper (Freudenberg H14C9, Fuel Cell Store) to reach a catalyst areal loading of ~0.3 mg/cm$^2$. The anode of IrO$_x$/Ti mesh (iridium oxide supported on titanium mesh) was prepared by a dip-coating and thermal-treatment method[57]. An anion-exchange membrane (Sustainion X37-50 Grade 60, Dioxide Materials Company) was adopted here, which needs activation in 1 M KOH for 48 h and washing with ultra-pure water prior to use. 0.25/0.5 M $K_2HPO_4$ electrolyte was used as the anolyte and was circulated using a peristaltic pump (F01A-STP, Kameor) at 10 mL/min. The flow rate of the CO gas flowing into the cathode flow field was kept at 40 SCCM by a mass flow controller (CS200A, Beijing Sevenstar Flow Co., Ltd.). CO was flowed through a humidifier prior to the MEA cathode flow field. Chronopotentiometry ($V$–$t$) measurements were conducted using a DH7003 potentiostat (from Jiangsu Donghua Analytical Instruments Co. Ltd.) at the current ranging from 10 to 100 mA/cm$^2$. The voltage was recorded without $iR$ correction. The cathode liquid-phase products were collected from the cathode outlet directly for 60 s with 1 mL of distilled water. The anode liquid-phase products were collected in the $K_2HPO_4$ anolyte at each $V$–$t$ run. The total FE of methanol was calculated based on the sum of the cathode and the anode liquid-phase products.

## Computational details
The $CO_2$/CO reduction reaction ($CO_2$RR/CORR) mechanism was investigated at the density functional theory (DFT) level of the Perdew–Burke–Ernzerhof hybrid functional (PBE0)[58] and def2-SVP[59,60] basis set using Gaussian 16 quantum chemical package[61]. The long-range van der Waals interactions were considered with Grimme's DFT-D3(BJ) empirical dispersion correction[62]. The minimum of studied geometries at the potential energy surface was optimized in the liquid solvent, in which water was regarded as an implicit solvent by the Solvation model density solvation model[63]. Vibrational frequency calculations further confirmed that the optimized structures had no imaginary frequency and provided the data of the infrared spectra, whose frequency scale factor is 0.9547[64], as well as the Gibbs free energies at 298.15 K and 1 atm. Notably, Gibbs free energies of the $CO_2$RR intermediates in electrochemical reaction pathways were calculated from the computational hydrogen electrode model proposed by Nørskov et al.[26,65]. The H$^+$/e$^-$ pair chemical potential value is half of the $H_2$ gas phase at 0 V and 1 atm.[66,67], whose reference is the RHE. The quantity of electron transfer was analyzed by Bader charge calculated by Multiwfn 3.8 (dev)[68].

## Data availability

The XAS data, product quantification, NMR spectra, in situ FTIR data, in situ XAS data, and theoretical calculations data generated in this study have been deposited in the Figshare database under accession code (https://doi.org/10.6084/m9.figshare.22722886)[69].

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

## Acknowledgements

This work was financially supported by the National Key Research and Development Program of China (No. 2021YFA1500502 (X.L.)), the National Natural Science Foundation of China (22102176 (X.L.) and 21925803 (Y.H.)), CAS Project for Young Scientists in Basic Research (YSBR-051 (X.L.)), the Strategic Priority Research Program of the Chinese Academy of Sciences (XDB36030200 (Y.H.)), the City University of Hong Kong start-up fund (B.L.). We thank the BL11B beamline at the Shanghai Synchrotron Radiation Facility (SSRF) and the BL17C beamline at National Synchrotron Radiation Research Center (NSRRC, Taiwan) for their help in performing the XAS measurements. We also thank the Photon Science Center for Carbon Neutrality of CAS for its support in instrument characterization. M.R. acknowledges the Institut Universitaire de France (IUF) for partial financial support.

## Author contributions

X.R., X.L., M.R., and B.L. designed and conceived the experiment. X.R., J.S., A.S., and C.L. performed the catalyst synthesis, structural characterization, and CORR electrocatalytic measurements. J.Z. and T.G. conducted the theoretical calculations. B.P. and Y.L. performed CORR MEA electrolysis. X.R. and J.D. performed the in situ electrochemical ATR-SEIRAS measurements and analyses. W.H., W.Z., and S.H. contributed to the in situ XAS spectroscopy measurements and analyses. X.R. and S.W. performed the in situ Raman spectroscopy measurements and analyses. T.G. and E.B. carried out DFT data analysis. X.R., X.Z., X.L., M.R., and B.L. wrote and revised the manuscript with inputs from all authors. The project was supervised by X.L., Y.H., M.R., and B.L.

## Competing interests

The authors declare no competing interests.
