## [Peer review file · Nature Communications]

REVIEWER COMMENTS

Reviewer #1 (Remarks to the Author):

The paper entitled “In-Situ Spectroscopic Probe of the Intrinsic Structure Feature of Single-Atom Center in Electrochemical CO/CO₂ Reduction to Methanol” by Ren, Y et al. provides experimental and theoretical considerations for understanding the CO₂-to-CO and CO-to-CH₃OH electrochemical reaction pathways by immobilizing molecular cobalt complexes on multi-walled carbon nanotubes. In-situ ATR-SEIRAS spectra, Raman, and K-edge XAS analyses are well combined with DFT calculations to provide a reasonable explanation. The critical conclusions based on these experimental and theoretical results are presented below:

1. On model single-Co-atom catalyst, the electrochemical reduction of CO to methanol occurs favorably, while the reduction pathway of CO₂ to methanol is strongly decreased.
2. The higher electron density of Co single atoms in CORR than in CO₂RR is the root cause of the different product selectivities.
3. The low energy barrier for forming a [H-CoPc-CO]⁻ species favors the methanol pathway.

The paper is well-written and provides a comprehensive understanding of single-atom catalysis through molecular catalysis. The conclusions of the experiments are an important addition to the current research on the catalytic mechanism of CoPc catalysts. As a result, the work is recommended for publication in Nature Communications when the following questions have been resolved.

Q1. During the performance tests carried out by the authors, the authors changed electrolytes several times. A 0.5 M K₂HPO₄ electrolyte was used in the H-Cell and a 0.25 M K₂HPO₄ electrolyte was used in the MEA tests. In the comparative CO₂ reduction test, a 0.5 M KHCO₃ electrolyte was used (Supplementary Fig. 10). The authors did not keep the cation concentration and anion species consistent, whereas recent work has shown that the composition of the electrolyte affects the activity of the reaction. The authors can give further explanation and clarification.

Q2. I observed that the authors have also calculated the CO vibrational frequency on the catalyst surface (Supplementary Fig. 18). Although it shows the same trend as the IR results, the specific values are not the same (Figure 3). Can the authors give some explanation for the difference in values?

Q3. In the MEA test, the liquid phase product may cross to the anode. The authors should give details of the method of collecting the liquid phase product (DOI: 10.1038/s41467-021-26053-w).

Q4. It is recommended that the authors provide a description of the method for obtaining the background and the reference spectra in IR tests (DOI: 10.1038/s41467-022-35450-8).

Q5. Raman spectroscopy shows that there are many vibrational signals of species, which may originate from electrolytes or from carbon nanotubes. Can the authors give details of the attribution?

Q6. In Figure 4c, the spectra in CO₂ and CO at open circuit potential (OCP) seem to be slightly different, what is the reason?

Q7. An additional explanation could be provided as to why H-CoPc functions as the stable structure for the reaction, maybe including more characterization data (like TEM) of the samples after electrochemical reactions.

Q8. How stable is the H-CoPc- species? Why the electron locates around the Pc ring, nor is the Co(II) center that proceeds to become Co(I)? The authors should present related DFT calculations on the stability for the comparison of H-Co(II)Pc- and H-Co(I)Pc.

Q9. What I really concern about is the H-CoPc species, which authors regard as the trigger molecule for CO to CH₃OH reaction. In numerous works of literature, CoPc is taken as the catalyst for CO₂RR, not the H-CoPc. Except for the experimental evidence, namely, H⁺ and one electron inflow into the CoPc and form H-CoPc, is there any theoretical calculation elucidating that H-CoPc is better than CoPc, for example, the CO₂ activation of H-CoPc is preferential over the CoPc, in other words, what is the difference and influence on the CO₂RR when considering H⁺ and one electron into CoPc molecule.

Q10. In the LSV results, a clear reduction peak is always observed in the atmosphere of CO (Supplementary Fig. 2), while it is difficult to be observed in the atmosphere of Ar and CO₂ (Supplementary Fig. 11), can the authors give a further explanation?

Q11. Could the authors give the gas phase product FE during stability testing (Supplementary Fig. 7)? It is suggested that the authors explain the reason for the step-like appearance of the current density.

Some minor problems:

1. P.6 L. 102, "that" should be removed.
2. The fonts of pictures in SI should be kept uniform.
3. The units of scan rate should be unified, the main text uses mV/s, while the SI uses mV s⁻¹ (such as Figure S2, S3).

Reviewer #2 (Remarks to the Author):

This is an intriguing paper to see CORR to MeOH while CO₂RR does not work for MeOH. While the mechanistic explanations are rather weak, I find this paper publishable in nature comm and give my comments below:

CORR is carried out in K₂HPO₄, would it be possible to test in similar conditions to CO₂RR? E.g. K_xHyCO₃ electrolyte at pH 7

Writing species as H-CoPc-CO is confusing to me. Is hydrogen on the backside of this motif or at the same side as CO ? or where is it?

Gibbs free energy diagram in Figure 5 is very limited, and there seems no further supplementary figure. Several questions arise and comparisons are needed for this to be valid: Is path to *COH checked? Is path to CH₄ checked? Is other MNC catalyst checked? Is comparison to Cu catalyst checked?

Overall, why should a path of CO-to-CHO lead to methanol? When is this path also suggest on Cu to lead to CH₄? It seems to me that CO-to-CHO is not they main descriptor for MeOH production.

And why is methanol limited to Co-based MNC? Not FeNC based?

I would highly value a comparison figure with other catalyst performance to MeOH!

Supplementary fig 16 for H-CoPc how do you get E = -3.9 eV?? Is a reference for H used?

Reviewer #3 (Remarks to the Author):

In this manuscript, Ren et al. report the preparation of a model single-Co-atom catalyst for high-performance electrochemical reduction of CO to methanol (CORR). The authors performed a variety of in-situ/operando spectroscopy experiments to elucidate the fundamental reasons behind the selectivity of CO (rather than CO₂) reduction to methanol. Both experimental and theoretical evidence suggest that the intrinsic structural feature of the single-Co-atom center, associated with different configurations of the *CO intermediate in CORR and CO₂RR, is the key factor. Overall, the corresponding results described in this paper are reasonable, the topic of single-atom electrocatalytic production of methanol is of great importance and will be of interest to the readership of Nature communication. However, before publication, some comments below are suggested to be considered.

1. The characterization of SACs is necessary to determine the properties of materials, and more detailed experimental evidence is needed. Therefore, in addition to the characterizations of HAADF-STEM and XAS, other auxiliary characterization results such as XRD, XPS... can be added to describe the properties of the catalyst more comprehensively.

2. It would be better to state the cobalt content in the catalyst, since cobalt is a possible active site. In addition, the influence of the MWCNT on the CO reduction activity should be ruled out.
3. In Fig. 5b, the H⁺ in the process from CoPc- to H-CoPc is unclear, should it be H₂O in and OH⁻ out?
4. The structure of CoPc- is unclear? Is it more accurate to express it as Co(I), since the process of Co(II) to Co(I) occurs first in CV curves.
5. In the DFT discussion section, "Additionally, a quite high energy barrier of 0.72 eV is required for the reductive conversion of H-CoPc-CO into [H-CoPc-CO]⁻ intermediate...", why focus on the energy barrier of H-CoPc-CO to [H-CoPc-CO]⁻ instead of discussing the energy barrier of the gas-phase CO activation step on H-CoPc, that is, H-CoPc+CO to [H-CoPc-CO]⁻?
6. In Fig. 5, the authors assume that the reaction path is *CO*CHO->*OCH₂->*OCH₃->*+CH₃OH, is there any experimental backup. If no, the path may be *CO*COH->*CHOH->*CH₂OH->*+CH₃OH process. Thus, more evidence for the proposed path is suggested to be provided.

Dear Reviewers,

Thank you for your kind consideration of our manuscript (NCOMMS-23-9584-T) entitled “*In-Situ* Spectroscopic Probe of the Intrinsic Structure Feature of Single-Atom Center in Electrochemical CO/CO₂ Reduction to Methanol”. We have carefully revised the manuscript based on the valuable comments and suggestions provided by the reviewers and all revisions have been marked in blue in the revised manuscript. The following list the point-to-point responses to the reviewers’ comments:

REVIEWER COMMENTS

Reviewer #1

The paper entitled “In-Situ Spectroscopic Probe of the Intrinsic Structure Feature of Single-Atom Center in Electrochemical CO/CO₂ Reduction to Methanol” by Ren, Y et al. provides experimental and theoretical considerations for understanding the CO₂-to-CO and CO-to-CH₃OH electrochemical reaction pathways by immobilizing molecular cobalt complexes on multi-walled carbon nanotubes. In-situ ATR-SEIRAS spectra, Raman, and K-edge XAS analyses are well combined with DFT calculations to provide a reasonable explanation. The critical conclusions based on these experimental and theoretical results are presented below:

1. On model single-Co-atom catalyst, the electrochemical reduction of CO to methanol occurs favorably, while the reduction pathway of CO₂ to methanol is strongly decreased.
2. The higher electron density of Co single atoms in CORR than in CO₂RR is the root cause of the different product selectivities.
3. The low energy barrier for forming a [H-CoPc-CO]⁻ species favors the methanol pathway.

The paper is well-written and provides a comprehensive understanding of single-atom catalysis through molecular catalysis. The conclusions of the experiments are an important addition to the current research on the catalytic mechanism of CoPc catalysts. As a result, the work is recommended for publication in Nature Communications when the following questions have been resolved.

Q1. During the performance tests carried out by the authors, the authors changed electrolytes several times. A 0.5 M K₂HPO₄ electrolyte was used in the H-Cell and a 0.25 M K₂HPO₄ electrolyte was used in the MEA tests. In the comparative CO₂ reduction test, a 0.5 M KHCO₃ electrolyte was used

(Supplementary Fig. 10). The authors did not keep the cation concentration and anion species consistent, whereas recent work has shown that the composition of the electrolyte affects the activity of the reaction. The authors can give further explanation and clarification.

Response: We appreciate the reviewer for the careful review and insightful comment. Actually, the CORR performance was examined in both 0.5 M K_2HPO_4 and 0.5 M KHCO_3 electrolyte. As shown in Figure R1, in 0.5 M KHCO_3 electrolyte, the Faradaic efficiency of methanol is 31.5%, which is inferior to the result in 0.5 M K_2HPO_4 and the current density is unstable and continues to decrease within 2400 s of the test. Although the Faradaic efficiency of methanol can reach up to 56.7% through extensive efforts to optimize the catalyst ink formulation and applied potential (Supplementary Fig. 11), the disadvantage of decreasing current density is still noticeable. As a side note, this Supplementary Fig. 11 has been performed in KHCO_3 electrolyte in order to be coherent with initial publications^{1,2}.

Figure R1. (a) Time-dependent total current density at -0.7 V vs. RHE in 0.5 M KHCO_3 electrolyte. (b) ^1H NMR spectrum of the liquid products measured in D_2O .

Note also that the competitive experiment (Supplementary Fig. 10) where an equimolar amount of dissolved ^{12}CO and $^{13}\text{CO}_2$ were used as reactants has been also done in 0.5 M KHCO_3 and gave similar results as in 0.5 M K_2HPO_4 , the result is shown in Figure R2.

Figure R2. (a) Controlled potential electrolysis of CoPc/MWCNT at -0.85 V vs. RHE in 0.5 M KHCO₃ under an almost equimolar amount of ¹³CO₂ (1.15 mM) and ¹²CO (0.95 mM) (gas mixture 96.5% ¹²CO + 3.5 % ¹³CO₂). (b) ¹H NMR spectra of the solution taken after 10 min of electrolysis, showing the sole formation of ¹²CH₃OH, with a Faradaic yield of 8.4%.

The CORR performance test in MEA cell was also evaluated in 0.5 M K₂HPO₄ electrolyte and the result is shown below (Figure R3) and added in Supplementary Fig. 6b of the revised manuscript. As shown, the maximum methanol Faradaic efficiency is 32.7% at the current density of 30 mA/cm². Compared with Fig. 2c, the 0.25 M K₂HPO₄ electrolyte is more favorable for methanol production and better inhibiting hydrogen evolution reaction. According to the reviewer's suggestion, these results and associated discussion have now been added into the revised manuscript.

Figure R3. FE_{CH_3OH} and FE_{H_2} at different current densities in 0.5 M K_2HPO_4 electrolyte in MEA cell.

Q2. I observed that the authors have also calculated the CO vibrational frequency on the catalyst surface (Supplementary Fig. 18). Although it shows the same trend as the IR results, the specific values are not the same (Figure 3). Can the authors give some explanation for the difference in values?

Response: We appreciate the reviewer for raising the valuable question. Actually, we have spent great efforts in simulating the IR CO vibrational frequency, however, the reason for the difference of vibrational frequencies between experiment and calculation might be quite complex. For instance, the H-CoPc-CO and H-CoPc-CO⁻ in DFT calculations were performed without considering an electric field to represent the electrochemical process. As for theoretical calculation, the method, basis set, structure, solvent model, *etc.* have a large influence on the computational results. In a word, the geometry and electric field may be the major factors for the difference between experimental result and calculation. Although the specific values are not the same, the trend of the CO vibrational frequency is well consistent with the *in-situ* IR results, which supports the conclusion of “a different adsorption configuration (bond length and angle) of *CO reaction intermediate in CORR as compared to that in CO₂RR, with a weaker stretching vibration of the C–O bond in the former case”.

Q3. In the MEA test, the liquid phase product may cross to the anode. The authors should give details of the method of collecting the liquid phase product (DOI: 10.1038/s41467-021-26053-w).

Response: We appreciate the reviewer for the valuable comment. The cathode liquid-phase products were collected from the cathode outlet directly for 60 s with 1 mL distilled water. The anode liquid-phase products were collected in the K_2HPO_4 anolyte at each V-t run. The total Faradaic efficiency of methanol was calculated based on the sum of the cathode and the anode liquid-phase products. The details of the method to collect the liquid phase products are now added to the “Methods” section in the revised manuscript.

Q4. It is recommended that the authors provide a description of the method for obtaining the background and the reference spectra in IR tests (DOI: 10.1038/s41467-022-35450-8).

Response: Thank you for your valuable suggestion. The background in IR test was taken at open

circuit potential in Ar-saturated 0.5 M K_2HPO_4 electrolyte and then the reference spectra under Ar atmosphere were collected from -0.3 V to -0.8 V vs. RHE with a step width and duration of 100 mV and 180 s, respectively. Afterwards, CO/CO_2 gas flow was purged into the electrolyte to saturation, and the background was collected again at open circuit potential. The description of the method has been added to the “Methods” section in the revised manuscript.

Q5. Raman spectroscopy shows that there are many vibrational signals of species, which may originate from electrolytes or from carbon nanotubes. Can the authors give details of the attribution?

Response: We thank the reviewer for the very helpful comment. We collected the Raman spectrum of pure MWCNT in 0.5 M K_2HPO_4 electrolyte. As shown in Figure R4, three characteristic peaks at 1590 cm^{-1} , 1350 cm^{-1} and 990 cm^{-1} were observed, corresponding to the G peak and D peak of carbon nanotubes, and the adsorption of HPO_4^{2-} ions over carbon nanotubes, respectively. Moreover, signature vibrational peaks in the CoPc/MWCNT spectrum are well consistent with the vibrational features of CoPc molecule, so in the *in-situ* Raman tests, the changes of the peaks on CoPc/MWCNT correspond to the contribution of CoPc. These results and associated discussion are now added into the revised Supplementary Information (Supplementary Fig. 14).

Figure R4. Raman spectra of CoPc/MWCNT, CoPc and MWCNT in 0.5 M K_2HPO_4 electrolyte.

Q6. In Figure 4c, the spectra in CO₂ and CO at open circuit potential (OCP) seem to be slightly different, what is the reason?

Response: We appreciate the reviewer for the careful review. The slight difference of the spectra in CO₂ and CO environment at OCP is due to the different pieces of carbon paper used for the measurements. Considering that the structure of CoPc changes to H-CoPc after applying a potential, freshly prepared catalyst-coated carbon paper was used when changing the reaction atmosphere. Due to the different pieces of carbon paper used for the measurements, the white-line part in the spectra under OCP did not completely coincide in CO₂ and CO environment. Therefore, to rule out this effect on the conclusion, we compared the absorption edge and white-line intensities of Co at both -0.5 V and -0.7 V vs. RHE. As shown in Fig. 4d & 4e, the absorption edge shifted toward lower energies and the drops in white-line intensities were more pronounced in CO atmosphere at both -0.5 V and -0.7 V vs. RHE, indicating that Co sites exhibited a lower oxidation state in CORR compared to that in CO₂RR.

Q7. An additional explanation could be provided as to why H-CoPc functions as the stable structure for the reaction, maybe including more characterization data (like TEM) of the samples after electrochemical reactions.

Response: Thank you for your valuable suggestion. We considered H-CoPc as the stable structure for the reaction from the following three aspects:

a) As shown in the Raman spectra in Figure R5, the peak at 750 cm⁻¹, which can be assigned to the stretching vibration of M–N bonds and C–N–C bridge bonds in CoPc³, cannot be fully recovered upon setting back the electrode to OCP, demonstrating that the vibration of the Pc macrocycle is altered during electroreduction. Then, we calcined the carbon paper after reaction at 300 °C under Ar atmosphere, and found that the peak could return to its original state, indicating that the structure of CoPc was slightly changed during electroreduction and hydrogenation of the Pc macrocycle might occur.

Figure R5. Raman spectra of CoPc/MWCNT at OCP, OCP-after and calcined under Ar atmosphere at 300 °C.

b) As shown in Supplementary Fig. 16, two redox peaks of CoPc/MWCNT appeared under Ar atmosphere, implying that before combining with CO₂/CO, CoPc underwent a two-electron reduction process, the first occurred on Co(II) to form Co(I), and the second occurred on the Pc macrocycle.

c) We tried to calculate the binding energies of CoPc-CO⁻, CoPc-CO²⁻ and H-CoPc-CO⁻; the results are shown below and H-CoPc-CO⁻ exhibits the lowest binding energy, indicating that the protonation of Pc can improve the stability of CO binding on the doubly reduced CoPc.

To clarify the reviewer's concern, the associated discussion is now added into the revised Supplementary Information (Supplementary Fig. 18, 19).

Figure R6. The binding energies (eV) of CoPc-CO⁻, CoPc-CO²⁻ and H-CoPc-CO⁻.

Q8. How stable is the H-CoPc⁻ species? Why the electron locates around the Pc ring, nor is the Co(II) center that proceeds to become Co(I)? The authors should present related DFT calculations on the stability for the comparison of H-Co(II)Pc⁻ and H-Co(I)Pc.

Response: We appreciate the reviewer for the valuable comments and suggestions. We calculated the single point energy of H-CoPc⁻ as shown in the figure below. Here, H-CoPc⁻ is the doubly reduced form of the starting CoPc after being protonated. The global charge of the molecule is -1, therefore it is written as H-CoPc⁻. As for the location of the electrons on the molecule, it is quite evident from the CV curve and DFT calculation as discussed in our reply to Q7 that the first e⁻ goes to the metal, so that Co is Co(I) and the second e⁻ goes to the ligand, so that the ligand gets an additional negative charge, which is compensated by the addition of a proton. Overall, the CoPc has been reduced with 2 electrons and 1 proton, and its charge is -1 since it was neutral in the initial state. Since the molecules in this situation do not possess the same number of electrons, it is not required to compare molecules with different global charges.

Figure R7. The relative single point energy (eV) for H-CoPc⁻.

Q9. What I really concern about is the H-CoPc species, which authors regard as the trigger molecule for CO to CH₃OH reaction. In numerous works of literature, CoPc is taken as the catalyst for CO₂RR, not the H-CoPc. Except for the experimental evidence, namely, H⁺ and one electron inflow into the CoPc and form H-CoPc, is there any theoretical calculation elucidating that H-CoPc is better than CoPc, for example, the CO₂ activation of H-CoPc is preferential over the CoPc, in other words, what is the difference and influence on the CO₂RR when considering H⁺ and one electron into CoPc

molecule.

Response: Thanks very much for your comments. The H-CoPc species as the trigger molecule for CO to CH₃OH reaction was supported by both the experimental and theoretical results (*in-situ* Raman, CV and DFT calculations as discussed in our reply to Q7). According to the reviewer's valuable suggestion, we have performed more theoretical calculations as below:

In fact, we have made a lot of attempts on the optimization of the CoPc-CO₂, but all the results show that the O atom in CO₂ interacts with the Co atom in CoPc, which is a physical adsorption on the CoPc. After an H⁺ and e⁻ are added on the CoPc (viz. H-CoPc), the C atom in CO₂ can directly bond with the Co atom. As shown in Figure R8(a), CO₂ is obviously activated by the H-CoPc molecule because the O-C-O angle becomes 135.23° (the O-C-O angle of stable CO₂ molecule is 180°). As for the CoPc molecule, the O-C-O angle is 179.80°, which is nearly linear and means that CO₂ is not activated by the CoPc. Compared to CoPc, the adsorption energy of CO₂ is lower over the H-CoPc, indicating that the adsorption and activation of CO₂ over H-CoPc is easier compared to that over CoPc. Moreover, the d_{z2} orbital gains insight into CO₂ activation from the perspective of electronic structure. As shown in Figure R8(b), the d_{z2} orbital is delocalized over Co and N atoms for CoPc, while condensed at the Co atom when H⁺ and one electron inflow into the CoPc, namely, the H atom breaks the orbital symmetry and makes d_{z2} orbital mainly localize at the Co atom. As a result, more electrons locate at the Co atom in the H-CoPc compared to CoPc, which is also proved by smaller Bader charge of the Co atom and higher d_{z2} orbital energy, further elucidating that CO₂ is preferred to be activated by the H-CoPc.

Figure R8. (a) The optimized structures of H-CoPc-CO₂ and CoPc-CO₂ and their adsorption energies (eV). (b) The d_{z^2} Kohn–Sham molecular orbital whose isovalue is 0.02, red and cyan colors denote the positive and negative orbital phases, $E_{d_{z^2}}$ is the orbital energy of d_{z^2} (eV) and q^{Bader} is the Bader atomic charge (e) for the Co atom of H-CoPc and CoPc.

Q10. In the LSV results, a clear reduction peak is always observed in the atmosphere of CO (Supplementary Fig. 2), while it is difficult to be observed in the atmosphere of Ar and CO₂ (Supplementary Fig. 11), can the authors give a further explanation?

Response: Thank you very much for your insightful question. In the atmosphere of CO, the observation of a reduction peak centered at ca. -0.53 V vs. RHE is associated with CO reduction. While in the atmosphere of Ar, only hydrogen evolution reaction occurs, therefore, this peak could not be observed. Instead, two peaks corresponding to the two-electron reduction process over CoPc are observed, the first reversible peak at around 0.20 V vs. RHE arises from the reduction of Co(II) to Co(I), while the second reduction peak at around -0.28 V vs. RHE corresponds to the delocalization of the charge obtained onto the macrocycle (Supplementary Fig. 16).

In CO₂ atmosphere, we also examined the LSV curve of CO₂RR in 0.5 M KHCO₃ and obtained the same result (as shown in Figure R9), which is most probably due to the efficient CO₂ reduction

catalysis over CoPc/MWCNT: the greatly increased current density covered the typical reduction peak of the CoPc. This phenomenon has also been reported in our previous work⁴ and a number of studies related to CO₂RR over CoPc catalysts⁵⁻⁸.

Figure R9. (a) LSV curves of CoPc/MWCNT recorded at a scan rate of 5 mV/s in CO₂-saturated 0.5 M KHCO₃ electrolyte. (b) LSV curves [Ren, X. *et al. Sci. China Chem.* **63**, 1727–1733 (2020)]⁴. (c) LSV curves of CoPc/CNT-MD acquired in Ar- and CO₂-saturated 0.5 M KHCO₃ electrolyte at a scanning rate of 5 mV/s [Wu, X. *et al. Adv. Funct. Mater.* **32**, 2107301 (2022)]⁵. (d) LSV curves of CoPc/NH₂-CNT, CoPc/OH-CNT, CoPc/COOH-CNT, and CoPc/CNT in CO₂-saturated 0.5 M KHCO₃ electrolyte [Li, H. *et al. Nano Res.* **15**, 3056–3064 (2022)]⁶. (e) LSV curves recorded at 20 mV/s in CO₂-saturated or Ar-saturated 0.5 M KHCO₃ solution for Co@Pc/C, Co/C, and Pc/C [He, C. *et al. Angew. Chem. Int. Ed.* **59**, 4914–4919 (2020)]⁷. (f) Cyclic voltammograms at 5 mV/s in 0.1 M KHCO₃ solution [Zhang, X. *et al. Nat. Commun.* **8**, 14675 (2017)]⁸.

Q11. Could the authors give the gas phase product FE during stability testing (Supplementary Fig. 7)? It is suggested that the authors explain the reason for the step-like appearance of the current density.

Response: We appreciate the reviewer for the valuable comments and suggestion. The H₂ FE during the stability test is provided below. As shown in Figure R10, the FE_{H₂} increases as the test time is extended, which is derived from the gradual changing of electrolyte's pH during the reaction. In order

to maintain a stable production of methanol, we therefore replaced the electrolyte every 2-3 hours throughout the stability test, and the FE_{H_2} would return to its initial value after each electrolyte replacement. Therefore, the current density shows a step shape.

Figure R10. Current density and H_2 Faradaic efficiency in the stability test of CoPc/MWCNT in CO-saturated 0.5 M K_2HPO_4 electrolyte at -0.70 V vs. RHE.

Some minor problems:

1. P.6 L. 102, “that” should be removed.
2. The fonts of pictures in SI should be kept uniform.
3. The units of scan rate should be unified, the main text uses mV/s, while the SI uses $mV\ s^{-1}$ (such as Figure S2, S3).

Response:

1. Thank you for your comment. The “that” has been removed.
2. Thank you for the kind remind. The fonts of the pictures in Supplementary Fig. 10 and 11 have been unified.
3. Thank you for your comment. The units of scan rate have been unified as mV/s.

Reviewer #2

This is an intriguing paper to see CORR to MeOH while CO₂RR does not work for MeOH. While the mechanistic explanations are rather weak, I find this paper publishable in nature comm and give my comments below:

Response: Thank you for your careful review and providing us with constructive comments and suggestions. According to your valuable comments and suggestions, we have supplemented more experimental data and DFT calculations to make the conclusions more solid and further elaborate the reaction mechanism. The following list the point-to-point responses to the reviewer's comments.

CORR is carried out in K₂HPO₄, would it be possible to test in similar conditions to CO₂RR? E.g. K_xH_yCO₃ electrolyte at pH 7.

Response: We appreciate the reviewer for the careful review and valuable suggestion. The CORR measurement in 0.5 M KHCO₃ electrolyte was also examined. The working electrode was prepared by dispersing 0.5 mg of CoPc/MWCNT catalyst in 960 μL of water-isopropyl alcohol solution (volume ratio of 1:1) with 40 μL Nafion. The catalyst ink was then painted onto a carbon paper (area = 1 cm²). Chronoamperometry test was performed on a CHI660E potentiostat in 35 mL 0.5 M KHCO₃ electrolyte, and the current-time response was recorded at -0.70 V vs. RHE in 2400 s with sampling time interval of 0.128 s at each potential. Figure R11(b) demonstrates that the Faradaic efficiency of methanol is 31.5%, which is lower than the result in 0.5 M K₂HPO₄, and the current density is unstable and keeps dropping within 2400 s of the test.

We also made a lot of efforts to optimize the catalyst ink formulation and applied potential (Supplementary Fig. 11). The CoPc/MWCNT loaded gas diffusion layer (GDL) was prepared by drop-casting the catalyst ink (20 mg MWCNT, 2 mg CoPc and 200 μL 5 wt.% Nafion solution were dispersed in 20 mL ethanol and sonicated for 30 min) onto the carbon paper with a PTFE mask. The mask had a 1 cm² surface square hole with 3 mm thickness, which allowed to receive approximately 100 μL of catalyst ink each time. After repeating the drop-casting 20 times, the ink deposited carbon paper was placed at room temperature and dried for 3 days in air to form the final 1 cm² GDL. Controlled potential electrolysis was performed at -0.85 V vs. RHE in 0.5 M KHCO₃. After 1 h electrolysis, these optimized conditions led to 56.7% FE_{CH₃OH}, but the current density still decreased with prolonging reaction.

Figure R11. (a) Time-dependent total current density at -0.7 V vs. RHE in 0.5 M KHCO_3 electrolyte. (b) ^1H NMR spectrum of the liquid products measured in D_2O .

Writing species as H-CoPc-CO is confusing to me. Is hydrogen on the backside of this motif or at the same side as CO? or where is it?

Response: We thank the reviewer for raising the question. In all DFT-optimized structures in Fig. 5, the H atoms are bonded to the N atoms on the Pc macrocycle, so we denote them as H-CoPc. As for H-CoPc-CO, the H atom is still bonded to the N atom on the Pc macrocycle, and after optimization, the *CO does not bind to Co in a straight line, but bends away from the H atom, as shown in Figure R12(d,e).

In fact, we would like to put the H atom bonding with the N atom to the left direction for uniformity. However, it looks troublesome for some molecules, e.g., in H-CoPc- CO_2^- , H-CoPc-COOH and H-CoPc-CHO in Figure R12(a,b,c), one may reckon that one O atom bonds with the Co atom with the view of the H atom at left. To more accurately understand the appearance of the entire molecule, the view of that H atom in both the left direction and backside has been added in the Supplementary Fig. 20, which is shown below:

Figure R12. The geometries of (a) H-CoPc-CO₂⁻, (b) H-CoPc-COOH and (c) H-CoPc-CHO whose view of the H atom on the backside (top) and in the left direction (bottom), (d) H-CoPc-CO and (e) H-CoPc-CO⁻.

Gibbs free energy diagram in Figure 5 is very limited, and there seems no further supplementary figure. Several questions arise and comparisons are needed for this to be valid: Is path to *COH checked? Is path to CH₄ checked? Is other MNC catalyst checked? Is comparison to Cu catalyst checked? Overall, why should a path of CO-to-CHO lead to methanol? When is this path also suggest on Cu to lead to CH₄? It seems to me that CO-to-CHO is not they main descriptor for MeOH production.

Response: Thanks for your insightful question. To address this concern, we have performed the DFT calculations for the *COH reaction path, viz. *CO⁻ → *COH → *CHOH → *CH₂OH → * + CH₃OH, whose optimized geometries and Gibbs free energy plot are shown below. The Gibbs free energy barrier is 0.83 eV from *CO⁻ to *COH, while 0.40 eV from *CO⁻ to *CHO, elucidating that the reaction path of CO-to-CHO is preferred. All these results and associated discussion are now added into the revised Supplementary Information.

Figure R13. The reaction path for the methanol formation via $*\text{COH}$, where the path is $*\text{CO}^- \rightarrow *\text{COH} \rightarrow *\text{CHOH} \rightarrow *\text{CH}_2\text{OH} \rightarrow * + \text{CH}_3\text{OH}$.

Figure R14. The Gibbs free energies (eV) for the reaction path of methanol formation via $*\text{COH}$, whose path is shown in Figure R13.

As for the CO-to- CH_4 path, there are two possible reaction paths: $*\text{CO}^- \rightarrow *\text{CHO} \rightarrow *\text{OCH}_2 \rightarrow *\text{OCH}_3 \rightarrow *\text{O} + \text{CH}_4 \rightarrow * + \text{H}_2\text{O}$ and $*\text{CO}^- \rightarrow *\text{COH} \rightarrow *\text{CHOH} \rightarrow *\text{CH}_2\text{OH} \rightarrow * + \text{CH}_4 + \text{H}_2\text{O}$,

respectively⁹. Therefore, we have calculated the Gibbs free energies of both CO-to-CHO and CO-to-COH reaction paths. As shown in Figure R14, the latter possesses a larger energy barrier than the former, thus, the *OCH₃ to *O + CH₄ path is calculated for the comparison of the CO-to-CH₃OH and CO-to-CH₄ paths (Figure R15, 16). According to the Gibbs free energies indicated in Figure R16, the energy barrier from *OCH₃ to CH₄ is 0.70 eV, while the formation of CH₃OH is an exothermic process (-0.37 eV), clearly elucidating the preference of CO-to-CH₃OH path.

According to the above findings, CO prefers to lead to methanol rather than methane via the CO-to-CHO path on CoNC. Similar DFT calculation paths have also been reported in the literature for methanol production over cobalt COF material¹⁰.

Figure R15. The reaction path for the conversion of CO to CH₄.

Figure R16. The Gibbs free energies (eV) for the CO-to-CH₃OH and CO-to-CH₄ paths.

In fact, the CORR performance over other MNC catalysts (including FePc, NiPc, and CuPc) were also tested. As shown in Figure R19, no CH₃OH production was observed over other MNC catalysts. According to the reviewer's valuable suggestion, further DFT calculation was performed and the Gibbs free energies from *CO⁻ to *CHO over FePc, NiPc, and CuPc are listed in Table R1 and Figure R17, the related structures are shown in Figure R18. It can be seen that all other metals have higher energy barriers from *CO⁻ to *CHO than H-CoPc. Hence, H-CoPc is favored to be used to gain CH₃OH from the theoretical point of view after comparison with other MNC catalysts.

Table R1. The Gibbs free energy barriers (eV) from *CO⁻ to *CHO on H-CoPc, FePc, H-FePc, NiPc, H-NiPc, CuPc, and H-CuPc.

Molecules	G _{barrier}
H-CoPc	0.402
FePc	0.663
H-FePc	0.662
NiPc	0.672
H-NiPc	0.767
CuPc	0.773

H-CuPc	0.699
--------	-------

Figure R17. Calculated Gibbs free energy diagrams of *CO⁻ and *CHO on H-CoPc, FePc, H-FePc, NiPc, H-NiPc, CuPc, and H-CuPc.

Figure R18. The geometries for *CO⁻ and *CHO on FePc, H-FePc, NiPc, H-NiPc, CuPc, and H-CuPc catalysts, the white, gray, blue, red, cyan, ice blue, and orange colors denote H, C, N, O, Fe, Ni, and Cu atoms, respectively.

As for the Cu-based catalysts, the activity and selectivity of bulk Cu towards producing methanol

are usually low¹¹⁻¹⁵, and the CH₃OH or CH₄ path over Cu catalysts is usually competitive. For instance, bimetallic alloys (Cu-Se¹⁶, Cu-Pd¹⁷), MOF-derived Cu@Cu₂O¹⁸, Cu₂NCN crystal¹⁹ were reported with relatively high methanol selectivity due to the sufficient grain boundaries, synergistic effect between Cu⁰ and Cu⁺, and highly delocalized electrons on Cu sites, respectively.

In detail, the pathway to CH₃OH or CH₄ formation involves the adsorption of *OCH₃ on the Cu surface to form Cu-*O-CH₃, and the subsequent breaking of the Cu-O or O-C bond, leading to either *OCH₃ or *CH₃ intermediate, finally producing CH₃OH or CH₄. Indeed, recent study has reported that in the electrocatalyst with isolated ionic Cu species, introducing suitable delocalized electron states may substantially reduce the Cu-O bond strength (compared with the O-C bond) in aqueous environment, converting the Cu-O bond cleavage and formation of *OCH₃ (and then CH₃OH) into an energetically favorable pathway¹⁹. This is consistent with our DFT calculations, which demonstrate that *OCH₃ to CH₃OH is more energetically favorable compared to the high energy barrier of *OCH₃ to CH₄ on CoPc.

According to the reviewer's valuable suggestion, we also synthesized MWCNT-supported CuPc catalyst and the catalytic performance for CH₃OH production in CORR was compared with the Co single-atom catalyst. As shown in Figure R19(d,e), no CH₃OH product could be detected in the electrolyte after CORR test. The causes for the inactivity of CuPc/MWCNT were also discussed based on DFT calculations (Table R1 and Figure R17), that is, the high energy barrier from *CO⁻ to *CHO restricts the hydrogenation of CO and the pathway to methanol.

These results and associated discussion have now been added into the revised manuscript for clarification.

And why is methanol limited to Co-based MNC? Not FeNC based?

I would highly value a comparison figure with other catalyst performance to MeOH!

Response: We thank the reviewer for raising the valuable question. As indicated in Figure R19 below, we tested the performance of various metal phthalocyanines for CORR. The results reveal that FePc, NiPc, and CuPc did not exhibit any CORR activity other than hydrogen evolution reaction, and the current densities are extremely low. No CH₃OH product could be detected in the electrolyte after the chronoamperometry test.

Figure R19. Time-dependent total current densities at different applied potentials for (a) FePc/MWCNT, (c) NiPc/MWCNT and (e) CuPc/MWCNT catalyst. ¹H NMR spectra of the liquid products were measured in D₂O and no liquid products were identifiable (b,d,f).

Additionally, we investigated the causes for the inactivity of other metal phthalocyanines using DFT calculations. As shown in Table R1 and Figure R17, compared with H-CoPc, other MNC catalysts have higher energy barriers from *CO⁻ to *CHO, which restricts the hydrogenation of CO and the pathway to methanol.

Supplementary fig 16 for H-CoPc how do you get $E = -3.9$ eV?? Is a reference for H used?

Response: We thank the reviewer for raising the question. The “ $E = -3.9$ eV” was obtained from the following expression,

$$E = E(\text{H-CoPc}) - E(\text{CoPc}) - 0.5E(\text{H}_2)$$

where the terms on the right side of the equation are the single point energies of H-CoPc, CoPc and H₂, respectively. The CoPc is regarded as the reference geometry, whose single point energy is set to 0 eV. Thus, the single point energy of H-CoPc relative to CoPc is calculated to be -3.9 eV.

To avoid ambiguity, we have revised the title of Supplementary Fig. 17 to “Relative single point energies of CoPc, CoPc⁻ and H-CoPc”.

Reviewer #3 (Remarks to the Author):

In this manuscript, Ren et al. report the preparation of a model single-Co-atom catalyst for high-performance electrochemical reduction of CO to methanol (CORR). The authors performed a variety of in-situ/operando spectroscopy experiments to elucidate the fundamental reasons behind the selectivity of CO (rather than CO₂) reduction to methanol. Both experimental and theoretical evidence suggest that the intrinsic structural feature of the single-Co-atom center, associated with different configurations of the *CO intermediate in CORR and CO₂RR, is the key factor. Overall, the corresponding results described in this paper are reasonable, the topic of single-atom electrocatalytic production of methanol is of great importance and will be of interest to the readership of Nature communication. However, before publication, some comments below are suggested to be considered.

1. The characterization of SACs is necessary to determine the properties of materials, and more detailed experimental evidence is needed. Therefore, in addition to the characterizations of HAADF-STEM and XAS, other auxiliary characterization results such as XRD, XPS... can be added to describe the properties of the catalyst more comprehensively.

Response: According to the reviewer's valuable suggestion, the XRD patterns and Co 2p XPS spectra for CoPc/MWCNT and CoPc (shown below) have been added to the revised Supplementary Information (Supplementary Fig. 2). As shown in Figure R20(a), no characteristic peaks associated with CoPc could be observed in the X-ray diffraction (XRD) pattern of CoPc/MWCNT, indicating that the CoPc molecules were uniformly dispersed. In addition, the binding energy of the Co 2p_{3/2} peak in the X-ray photoelectron spectroscopy (XPS) spectrum was located at 780.7 eV, corresponding to the +2 valence state of Co in Co-N coordination (Figure R20(b)).

Figure R20. (a) XRD patterns of CoPc/MWCNT, MWCNT and CoPc. (b) Co 2p XPS spectra for CoPc/MWCNT and CoPc.

2. It would be better to state the cobalt content in the catalyst, since cobalt is a possible active site. In addition, the influence of the MWCNT on the CO reduction activity should be ruled out.

Response: We appreciate the reviewer for the valuable suggestions. The cobalt content in the single-Co-atom catalyst is 0.85 wt.% based on inductively coupled plasma optical emission spectroscopy (ICP-OES) measurements. The corresponding result has been added in the revised Supplementary Information.

We have performed the electrochemical reduction experiment of MWCNT, however it is inactive for CO electroreduction. As shown in Figure R21, in CO atmosphere, the current density of the MWCNT is extremely small ($< 1 \text{ mA/cm}^2$). No CH_3OH could be detected in the electrolyte collected from the chronoamperometry test at -0.70 to -0.78 V vs. RHE for 1 h.

Figure R21. (a) Time-dependent total current densities at -0.70 to -0.78 V vs. RHE of MWCNT. (b) ^1H NMR spectrum of the liquid products measured in D_2O and no liquid products were identifiable.

3. In Fig. 5b, the H^+ in the process from CoPc^- to H-CoPc is unclear, should it be H_2O in and OH^- out?

Response: Thank you for your comment. We have made the modification in Fig. 5b.

4. The structure of CoPc^- is unclear? Is it more accurate to express it as Co(I) , since the process of Co(II) to Co(I) occurs first in CV curves.

Response: Thank you very much for your kind remind. In the first step of the mechanism, one electron is transferred to CoPc (neutral) to generate the negatively charged species CoPc^- , in which the electron is located on the Co, so that Co is Co(I) . The global charge of the molecule is -1 since it was neutral in the initial state, therefore it is written as CoPc^- .

5. In the DFT discussion section, "Additionally, a quite high energy barrier of 0.72 eV is required for the reductive conversion of H-CoPc-CO into $[\text{H-CoPc-CO}]^-$ intermediate...", why focus on the energy barrier of H-CoPc-CO to $[\text{H-CoPc-CO}]^-$ instead of discussing the energy barrier of the gas-phase CO activation step on H-CoPc , that is, $\text{H-CoPc} + \text{CO}$ to $[\text{H-CoPc-CO}]^-$?

Response: Thank you for the question. We focus on the energy barrier of H-CoPc-CO to $[\text{H-CoPc-CO}]^-$ for two reasons:

a) As shown in Figure R22(a), in the CO_2 -to- CO path, the energy barrier from H-CoPc-CO to H-

CoPc+CO (0.38 eV) is lower than the reductive conversion of H-CoPc-CO into [H-CoPc-CO]⁻ intermediate (0.72 eV), making CO desorption easier to occur on H-CoPc and indicating the very unlikely possibility of H-CoPc-CO conversion to [H-CoPc-CO]⁻. When CO is desorbed, the lower energy barrier for CO₂ adsorption on H-CoPc⁻ (0.19 eV) compared to the CO adsorption (0.29 eV) makes CO₂ reduction occurs preferentially (Figure R22(b)). Therefore, the desorbed CO cannot be further reduced directly on H-CoPc, which is also the reason why the methanol pathway is largely suppressed in the CO₂/CO mixture.

Figure R22. (a) Calculated Gibbs free energy diagrams of CO₂-to-CO path. (b) Comparison of the calculated Gibbs free energy of the CO₂-to-CO and CO-to-MeOH paths.

b) In the CO-to-MeOH path, the activation of CO to form [H-CoPc-CO]⁻ must occur on H-CoPc⁻ rather than H-CoPc because the peak of CO reduction (-0.53 V vs. RHE) appeared at a more negative potential than the second reduction peak at around -0.28 V vs. RHE, which corresponds to the delocalization of the charge obtained onto the macrocycle in the cyclic voltammetry curve as shown in Supplementary Fig. 16.

6. In Fig. 5, the authors assume that the reaction path is *CO*CHO->*OCH₂->*OCH₃->*+CH₃OH, is there any experimental backup. If no, the path may be *CO*COH->*CHOH->*CH₂OH->*+CH₃OH process. Thus, more evidence for the proposed path is suggested to be provided.

Response: Thanks for your good suggestion about the reaction path. In light of your consideration, we have calculated the other possible reaction path, viz. *CO⁻ → *COH → *CHOH → *CH₂OH → * + CH₃OH, the related optimized geometries and Gibbs free energies are shown in Figures R23 and R24.

It can be seen that the energy barrier is 0.83 eV from $^*\text{CO}^-$ to $^*\text{COH}$, which is larger than the $^*\text{CO}^-$ to $^*\text{CHO}$ (0.40 eV). Thus, compared with the reaction path as shown in Fig. 5, the new path is more difficult to proceed due to a higher energy barrier.

Figure R23. The reaction path for methanol formation via $^*\text{COH}$, where the path is $^*\text{CO}^- \rightarrow ^*\text{COH} \rightarrow ^*\text{CHOH} \rightarrow ^*\text{CH}_2\text{OH} \rightarrow ^* + \text{CH}_3\text{OH}$.

Figure R24. The Gibbs free energies (eV) for the reaction path of methanol formation via $^*\text{COH}$,

whose path is shown in Figure R23.

Once again, thank you very much for your valuable comments and suggestions.

Sincerely yours,

Bin Liu

Professor

Department of Materials Science and Engineering

City University of Hong Kong

Email: bliu48@cityu.edu.hk

References

1. Boutin, E. *et al.* Aqueous electrochemical reduction of carbon dioxide and carbon monoxide into methanol with cobalt phthalocyanine. *Angew. Chem. Int. Ed.* **58**, 16172–16176 (2019).
2. Wu, Y., Jiang, Z., Lu, X., Liang, Y. & Wang, H. Domino electroreduction of CO₂ to methanol on a molecular catalyst. *Nature* **575**, 639–642 (2019).
3. Liu, Z., Zhang, X., Zhang, Y. & Jiang, J. Theoretical investigation of the molecular, electronic structures and vibrational spectra of a series of first transition metal phthalocyanines. *Spectrochim. Acta, Part A* **67**, 1232–1246 (2007).
4. Ren, X. *et al.* Electron-withdrawing functional ligand promotes CO₂ reduction catalysis in single atom catalyst. *Sci. China Chem.* **63**, 1727–1733 (2020).
5. Wu, X. *et al.* Molecularly dispersed cobalt phthalocyanine mediates selective and durable CO₂ reduction in a membrane flow cell. *Adv. Funct. Mater.* **32**, 2107301 (2022).
6. Li, H. *et al.* Coordination engineering of cobalt phthalocyanine by functionalized carbon nanotube for efficient and highly stable carbon dioxide reduction at high current density. *Nano Res.* **15**, 3056–3064 (2022).
7. He, C. *et al.* Molecular evidence for metallic cobalt boosting CO₂ electroreduction on pyridinic nitrogen. *Angew. Chem. Int. Ed.* **59**, 4914–4919 (2020).

8. Zhang, X. *et al.* Highly selective and active CO₂ reduction electrocatalysts based on cobalt phthalocyanine/carbon nanotube hybrid structures. *Nat. Commun.* **8**, 14675 (2017).
9. Zhang, J., Cai, W., Hu, F. X., Yang, H. & Liu, B. Recent advances in single atom catalysts for the electrochemical carbon dioxide reduction reaction. *Chem. Sci.* **12**, 6800–6819 (2021).
10. Yao, C.-L., Li, J.-C., Gao, W. & Jiang, Q. An integrated design with new metal-functionalized covalent organic frameworks for the effective electroreduction of CO₂. *Chem. Eur. J.* **24**, 11051–11058 (2018).
11. Albo, J. *et al.* Copper-based metal–organic porous materials for CO₂ electrocatalytic reduction to alcohols. *ChemSusChem* **10**, 1100–1109 (2017).
12. Le, M. *et al.* Electrochemical reduction of CO₂ to CH₃OH at copper oxide surfaces. *J. Electrochem. Soc.* **158**, E45 (2011).
13. Kuhl, K. P. *et al.* Electrocatalytic conversion of carbon dioxide to methane and methanol on transition metal surfaces. *J. Am. Chem. Soc.* **136**, 14107–14113 (2014).
14. Nitopi, S. *et al.* Progress and perspectives of electrochemical CO₂ reduction on copper in aqueous electrolyte. *Chem. Rev.* **119**, 7610–7672 (2019).
15. Kuhl, K. P., Cave, E. R., Abram, D. N. & Jaramillo, T. F. New insights into the electrochemical reduction of carbon dioxide on metallic copper surfaces. *Energy Environ. Sci.* **5**, 7050–7059 (2012).
16. Yang, D. *et al.* Selective electroreduction of carbon dioxide to methanol on copper selenide nanocatalysts. *Nat. Commun.* **10**, 677 (2019).
17. Lu, L. *et al.* Highly efficient electroreduction of CO₂ to methanol on palladium–copper bimetallic aerogels. *Angew. Chem. Int. Ed.* **57**, 14149–14153 (2018).
18. Yang, X. *et al.* MOF-derived Cu@Cu₂O heterogeneous electrocatalyst with moderate intermediates adsorption for highly selective reduction of CO₂ to methanol. *Chem. Eng. J.* **431**, 134171 (2022).
19. Kong, S. *et al.* Delocalization state-induced selective bond breaking for efficient methanol electrosynthesis from CO₂. *Nat. Catal.* **6**, 6–15 (2023).

REVIEWERS' COMMENTS

Reviewer #1 (Remarks to the Author):

The authors have done a great job in improving the paper and I am very happy to see the work they have done has added to the strength of their conclusions. I have no extra points and wish the authors the best of luck with their future experiments.

Reviewer #2 (Remarks to the Author):

The authors have thoroughly replied the reviewer questions. And while I trust the experimental results, I am still skeptical about the explanation of why the H-CoPC can make methanol.

I now understand that H-CoPC is stable with -3.9 eV with respect to CoPC given the authors reply.

This value means that one H binds very strongly to the outer N as shown in the pictures. However, there are 4N where H could bind. Have the authors tested 1,2,3,4 H at the N positions. I would think that the active motif might be H4-CoPC given the strong H-CoPC binding.

The authors do give a lengthy reply on why H-CoPC makes methanol. Given that it is really unclear why this is, and why metal catalyst does not make methanol, I think perhaps it is helpful to write it more open in the manuscript and incorporate parts of their reviewer reply in their manuscript.

That said, the paper is of interest to the readership of Nature communication.

Reviewer #3 (Remarks to the Author):

The authors have well addressed the concerns and the present manuscript should be accepted.

Dear Reviewers,

Thank you for spending time reviewing our manuscript (NCOMMS-23-09584A) entitled “*In-Situ* Spectroscopic Probe of the Intrinsic Structure Feature of Single-Atom Center in Electrochemical CO/CO₂ Reduction to Methanol”. The manuscript has been revised carefully according to the reviewers’ and editor’s valuable comments. We believe that the manuscript has been greatly improved and hope it has reached the standard of “*Nature Communications*”. Please see below, for a point-by-point response to the reviewers’ comments and concerns, and all revisions have been marked in blue in the revised manuscript.

REVIEWER COMMENTS

Reviewer #2

The authors have thoroughly replied the reviewer questions. And while I trust the experimental results, I am still skeptical about the explanation of why the H-CoPC can make methanol. I now understand that H-CoPC is stable with -3.9 eV with respect to CoPC given the authors reply. This value means that one H binds very strongly to the outer N as shown in the pictures. However, there are 4N where H could bind. Have the authors tested 1,2,3,4 H at the N positions. I would think that the active motif might be H₄-CoPC given the strong H-CoPC binding.

Response: Thank you for your careful review and constructive comments. According to your valuable comments, we have performed additional DFT calculations on 2,3,4 H bonded CoPc. Actually, the mechanism we proposed has already been suggested by Kaneko et al. in 1996 (J. Mol. Catal. A-Chem., 1996, 112(1): 55-61.), it was proposed that one-electron reduced CoPc species would be too weak to form an intermediate with CO molecule, while two-electron reduced species was necessary.

In our work, two redox peaks of CoPc/MWCNT was observed in cyclic voltammetry curves (Supplementary Fig. 16), the first reversible peak at around 0.20 V vs. RHE arises from the reduction of Co(II) to Co(I), while the second reduction peak at around -0.28 V vs. RHE corresponds to the delocalization of the charge obtained onto the macrocycle. In addition, DFT calculation results of the relative single point energy show that the CoPc can be easily protonated at the ligand (denoted as H-CoPc), making it more thermodynamically stable (Supplementary Fig. 17), which is also confirmed by the peak at 750 cm⁻¹ in the *in-situ* Raman spectra. Therefore, H-CoPc should be

reasonable to perform CO reduction to CH₃OH.

According to the reviewer's valuable suggestion, additional DFT calculations were performed. The optimized structure of H_x-CoPc (x=1-4) is shown in Figures R1 and R2. The relative energies show that H₄-CoPc (-3.355 eV) exhibits a stronger H-N binding than the H₂-CoPc and H₃-CoPc, but weaker than the H-CoPc (-3.976 eV), elucidating that the H-CoPc is more stable than the H₄-CoPc. Moreover, the results presented in Figure R2 confirm the lowest binding energy for H₄-CoPc-CO⁻ (-0.993 eV), indicating the stronger interaction between the H₄-CoPc⁻ and CO compared to that of H₁-CoPc (-0.687 eV, shown in Figure S19). However, it is notable that the H-CoPc is much more stable than the H₄-CoPc as shown in Figure S17 and Figure R1, and it needs to gain additional energy (~1.2 eV) to form H₂-CoPc and much more energy to form H₄-CoPc. Thus, the H-CoPc should be reasonable for the proposed mechanism for CO reduction to CH₃OH, which is confirmed by multiple evidences such as cyclic voltammetry curves, *in-situ* Raman spectroscopy, and DFT calculations in this work, and is also consistent with the conclusion proposed by Kaneko et al.

Figure R1. Relative single point energies of *o*-H₂-CoPc, *p*-H₂-CoPc, H₃-CoPc, and H₄-CoPc. The CoPc is regarded as the reference, whose single point energy is set to 0 eV. The energy for H_x-CoPc (x=1-4) relative to CoPc was calculated from $E = E(\text{H}_x\text{-CoPc}) - E(\text{CoPc}) - 0.5E(x\text{H}_2)$, where the terms on the right side of the equation are the single point energies of H_x-CoPc, CoPc, and H₂, respectively. The calculation results show that H-CoPc exhibits the lowest relative single point energy, indicating that CoPc is easily protonated at the ligand, making it more thermodynamically stable.

Figure R2. The binding energies of *o*-H₂-CoPc-CO⁻, *p*-H₂-CoPc-CO⁻, H₃-CoPc-CO⁻, and H₄-CoPc-CO⁻.

The authors do give a lengthily reply on why H-CoPC makes methanol. Given that it is really unclear why this is, and why metal catalyst does not make methanol, I think perhaps it is helpful to write it more open in the manuscript and incorporate parts of their reviewer reply in their manuscript.

Response: We appreciate the reviewer for the valuable suggestion. The discussion on metal catalysts that do not make methanol, based on experimental and theoretical evidences (Supplementary Fig. 24,25), have been added into the revised manuscript.

Once again, thank you very much for your valuable comments and suggestions.

Sincerely yours,

Bin Liu
 Professor
 Department of Materials Science and Engineering
 City University of Hong Kong
 Email: bliu48@cityu.edu.hk